# SurveillanceVQA-589K: A Benchmark for Comprehensive Surveillance Video-Language Understanding with Large Models

## Abstract

Understanding surveillance video content remains a critical yet underexplored challenge in vision–language research, particularly due to its real-world complexity, irregular event dynamics, and safety-critical implications. In this work, we introduce SurveillanceVQA-589K, the largest open-ended video question answering (VQA) benchmark tailored to the surveillance domain. The dataset comprises 589,380 QA pairs spanning 12 cognitively diverse question types, including temporal reasoning, causal inference, spatial understanding, and anomaly interpretation, across both normal and abnormal video scenarios. To construct the benchmark at scale, we design a hybrid annotation pipeline that combines temporally aligned human-written captions with Large Vision-Language Model (LVLM) assisted QA generation using prompt-based techniques. We also propose a multi-dimensional evaluation protocol to assess contextual, temporal, and causal comprehension. We evaluate 12 LVLMs under this framework, revealing significant performance gaps, especially in causal and anomaly-related tasks, underscoring the limitations of current models in real-world surveillance contexts. Our benchmark provides a practical and comprehensive resource for advancing video-language understanding in safety-critical applications such as intelligent monitoring, incident analysis, and autonomous decision-making. The dataset is publicly available at: `https://anonymous.4open.science/r/SurveillanceVQA-589K`.

## 1 Introduction

Surveillance videos have become pivotal data sources for smart cities (Kashef et al., 2021). In contrast to publicly available or social media videos, surveillance footage varies significantly in acquisition methods, content attributes, and application purposes (Xu et al., 2016; Tsakanikas & Dagiuklas, 2018). They encompass diverse spatiotemporal conditions (Nawaratne et al., 2020; Sreenu & Durai, 2019), spanning day-night cycles, varied weather, and heterogeneous environments like streets (Liang et al., 2023; Ma et al., 2019), shopping centers (Arroyo et al., 2015), and trans-

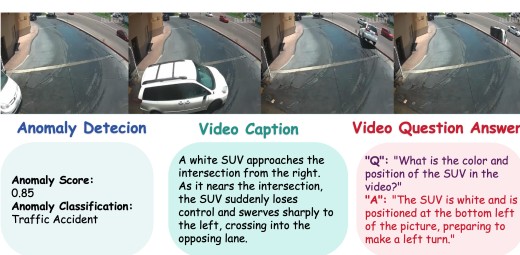

Figure 1: Anomaly detection, video caption vs. our VQA for surveillance applications.

portation hubs (Ling et al., 2017), resulting in high data heterogeneity. Abnormal events captured are often sudden, low-frequency, and diverse, posing challenges for perception and modeling (Liu et al., 2024a; 2023b).

Current computer vision tasks for surveillance videos primarily focus on the detection and classification of abnormal events, often relying on predefined event types and handcrafted features (Pawar & Attar, 2019; Zhou et al., 2019; Doshi & Yilmaz, 2020; Al-Lahham et al., 2024; Zanella et al., 2024; Wu et al., 2024). While such goal-oriented approaches can be effective in specific scenarios, they typically lack deeper semantic modeling of event progression, behavioral motivations, and environ-

mental context (Yuan et al., 2024a). This limits the potential of surveillance videos in intelligent urban governance, behavior prediction, and multimodal reasoning (Pathirannahalage et al., 2024).

To address these limitations, UCA (Yuan et al., 2024b;a), has proposed a multimodal understanding framework for surveillance videos. It incorporates fine-grained language annotations and temporal markers, covering tasks such as moment localization, caption generation, and dense captioning. However, UCA primarily focuses on descriptive tasks and lacks interactive question answering (QA) mechanisms, which makes it less aligned with recent trends in complex semantic understanding and reasoning within multimodal systems (Kim et al., 2025).

To further enhance the semantic reasoning capabilities of models in the surveillance domain, we introduce QA tasks to enable interactive and cognitively rich understanding, as shown in Figure 1. Unlike descriptive tasks, QA tasks allow models to perform logical reasoning, causal inference, and complex semantic analysis on video content. To achieve this, we construct SurveillanceVQA-589K, a large-scale QA dataset specifically designed for surveillance videos. The dataset consists of four surveillance video datasets as video sources and contains approximately 589,000 question-answer pairs, including 12 QA types covering both normal and abnormal video content, such as factual summarization, behavior/spatial-temporal analysis, causal reasoning, anomaly detection, etc. This design aims to elevate video understanding to a higher cognitive level.

We benchmark 8 local-deployed Large Vision-Language Models (LVLMs) on SurveillanceVQA-589K, including variants like VideoLLaMa3 (Zhang et al., 2025), LLaVA series (Zhang et al., 2024c;b; Li et al., 2024a), Qwen2.5-VL series (Bai et al., 2025), and InternVL series (Chen et al., 2024c), ranging from lightweight 0.5B to general-purpose 7B models. We also test the model performance of 4 API-called LVLMs (e.g., Gemini 2.5 Pro Google (2025), OpenAI's GPT-4o OpenAI (2024), Baidu's ERNIE 4.5 Turbo VL Baidu (2025), and the newest model InternVL-3.5 Wang et al. (2025)) on our abnormal videos. Despite their success in open-domain tasks, current LVLMs demonstrate clear limitations in surveillance scenarios. Performance on complex tasks, such as causal inference and abnormal event analysis, remains poor, with most models scoring below the midpoint. Furthermore, although fine-tuning on local-deployed LVLMs enhances general understanding, it does not lead to substantial improvements in complex reasoning tasks. Through a visualization of failure cases, we analyze the underlying causes in detail and provide suggestions to guide future model development. Our benchmark reveals systematic weaknesses in causal inference and anomaly understanding, offering a practical testbed for real-world surveillance applications.

## 2 RELATED WORK

### 2.1 SURVEILLANCE VIDEO ANALYSIS BENCHMARK

In recent years, there has been a growing interest in the academic community toward understanding the content of surveillance videos, which has led to the development of several benchmark datasets to support research in this domain. Early datasets predominantly focused on anomaly detection in surveillance scenarios, including UCSD Ped1 and Ped2 (Li et al., 2013), the Avenue dataset (Lu et al., 2013), the Subway dataset (Adam et al., 2008), the ShanghaiTech Campus dataset (Luo et al., 2017), UCF-Crime (Sultani et al., 2018), MEVA (Corona et al., 2021), NWPU Campus dataset (Cao & Others, 2023), and MSAD (Zhu et al., 2024). While these datasets have laid a strong foundation for surveillance video analysis, the majority of these datasets still lack accompanying textual annotations, rendering them inadequate for comprehensive multimodal vision-language research. Notably, UCA (Yuan et al., 2024b) distinguished itself from prior surveillance datasets through its rich linguistic annotations. However, the current version of UCA includes only approximately 20,000 manually labeled descriptions and lacks an interactive, question-answering (QA) based evaluation framework. Such a framework is increasingly recognized as a critical tool for assessing high-level reasoning, anomaly understanding, and semantic generalization in modern multimodal systems.

To address this gap, we propose the construction of a novel QA-driven benchmark for surveillance video understanding. This benchmark is designed to enable interactive evaluation and foster deeper semantic reasoning over real-world surveillance video content.

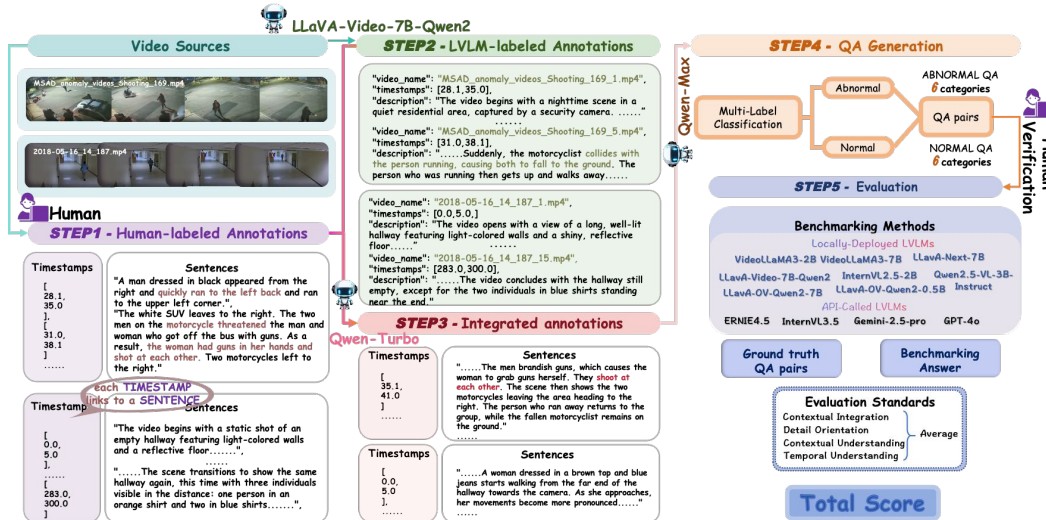

Figure 2: Our overall framework, including QA generation and evaluation.

## 2.2 Video-language Understanding Benchmark

With the emergence of LVLMs (Li & Lu, 2024; Cui et al., 2023; Liu et al., 2023a), traditional benchmarks have become increasingly inadequate in capturing the full range of model capabilities. Recent benchmarks aim to address this by incorporating diverse tasks (Wang et al., 2024b; Li et al., 2025), multi-level granularity (Wang et al., 2024a), and scalable evaluation protocols (Chen et al., 2024a). Benchmarks such as OwlEval, MME (Fu et al., 2023), SEED-Bench (Li et al., 2024b), MM-Vet (Yu et al., 2023), and MMBench (Liu et al., 2024b), MVBench (Li et al., 2024c), Vision-R1 (Huang et al., 2025), and EMMA (Hao et al., 2025) cover a wide spectrum of tasks, from image captioning and reasoning to fact verification. These works propose multi-dimensional metrics—including linguistic consistency, semantic alignment, and visual grounding—to characterize model behavior more comprehensively. In the video domain, models must handle temporal dynamics and evolving semantics (Weng et al., 2024; Chen et al., 2024b). Earlier benchmarks such as TVQA (Lei et al., 2018) and Next-QA (Xiao et al., 2021) used multiple-choice formats to assess temporal localization and event comprehension. More recent work, such as VideoChatGPT (Maaz et al., 2023), introduces multi-turn QA grounded in video input, emphasizing coherence and contextual consistency over extended interactions. FunQA (Xie et al., 2024) pushes reasoning further by evaluating a model's ability to identify unexpected or humorous events, highlighting challenges in modeling incongruity and causal anomalies in temporal sequences. Svbench (Yang et al., 2025) proposes temporal multi-turn QA chains, which are specifically for streaming video understanding.

Our work extends these efforts by addressing the unique challenges of surveillance video analysis. Unlike open-domain or entertainment-based datasets, SurveillanceVQA-589K emphasizes real-world diverse abnormal events, spatiotemporal analysis, complex reasoning, etc.

## 3 SURVEILLANCEVQA-589K

The SurveillanceVQA-589K dataset includes 31,548 video clips with textual annotations, 27,962 clips labeled as normal and 3,586 as anomalous, resulting in a total of 589,380 QA pairs. The following contents show the procedure of QA pairs generation, illustrated in Figure 2.

### 3.1 VIDEO ANNOTATION GENERATION

Following the annotation protocol established in UCA (Yuan et al., 2024b), we extended manual annotation efforts to the MSAD, MEVA, and NWPU surveillance datasets. This process involved generating event-level captions that included both precise timestamps and detailed event descriptions. The detailed annotation procedures are shown in Appendix A.

Subsequently, using the timestamp information obtained during the manual annotation phase, we employed the video processing toolkit MoviePy to automatically segment the original videos and extract the corresponding short clips. We then utilized the powerful multimodal model LLaVA-Video-7B (Zhang et al., 2024c) to perform in-depth analysis on each segmented clip, generating detailed descriptions. As a result, we obtained 31,548 segment-level annotations produced by the LVLM.

Then, we performed a deep integration of the high-quality information obtained from manual annotations with the rich segment-level descriptions generated by the large multimodal model LLaVA-Video. The objective of this integration was to combine the precision of human annotations with the diversity and depth of model-generated descriptions. The guiding principle for prompt design was to preserve the semantic

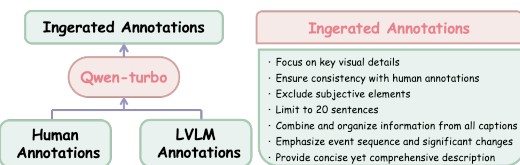

Figure 3: Procedure of integrating annotations.

integrity of human annotations while enriching them with complementary information from the model output. To facilitate this process, we introduced Qwen-Turbo (Yang et al., 2024), an advanced language model, to leverage its natural language processing capabilities for comprehensive analysis, redundancy elimination, and content optimization. Specifically, Qwen-Turbo was tasked with identifying and resolving redundant or inconsistent expressions, while enhancing semantic richness and logical coherence. This resulted in more fluent, structured, and contextually aligned event-level descriptions for each video clip.

Through this integration, we obtained 31,548 refined textual annotations. The procedure for integrating human and LVLM annotations, along with the generation prompt, is illustrated in Figure 3. The examples of human-labeled, LVLM-labeled, integrated annotations are shown in Appendix A.

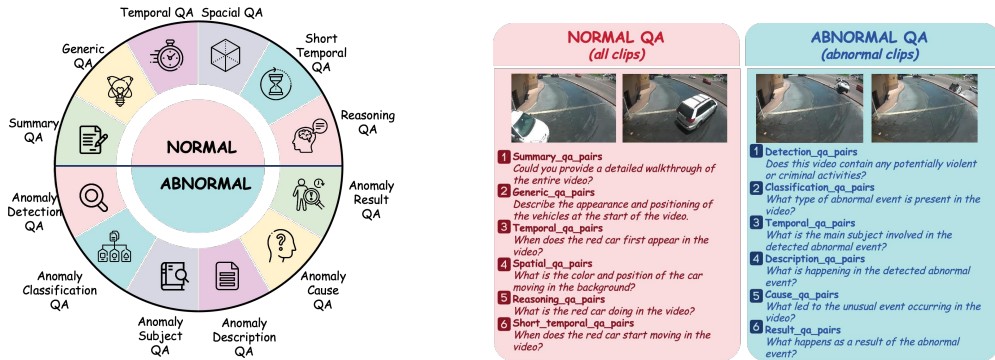

(a) The involved 12 QA categories.    (b) The example questions of each QA task.

Figure 4: QA types and example questions of our SurveillanceVQA-589K.

## 3.2 AUTOMATIC QA GENERATION

After obtaining the clip-level descriptions, we proceeded to generate QA pairs. Specifically, we employed Qwen-Max (Yang et al., 2024) to analyze the annotation data and classify each video segment as either normal or abnormal.

After obtaining two distinct sets of video clips (normal vs. abnormal) along with their corresponding textual annotations, we designed category-specific prompts to guide the generation of QA pairs. For normal clips, the prompts were crafted to elicit a comprehensive understanding of video content, focusing on global scene descriptions, temporal sequencing, spatial detail extraction, and behavioral inference. These prompts support general video comprehension and open-ended QA tasks. In contrast, for abnormal clips, the prompts emphasized event detection, anomaly type classification, subject identification, detailed incident descriptions, and causal reasoning. This design is tailored to the specific requirements of surveillance anomaly detection and incident-level semantic analysis.

Based on the customized prompts, we employed Qwen-Max to generate QA pairs for each video clip, following different generation strategies for normal and abnormal categories. For normal QA

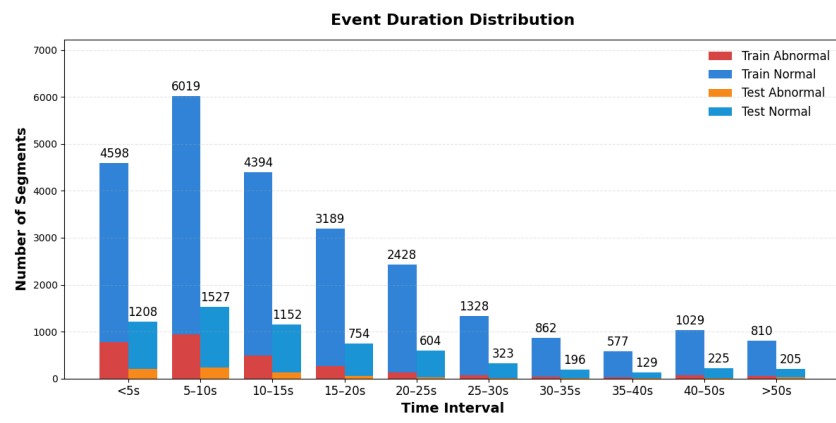

Figure 5: Distribution of event durations on the training/test sets

tasks, six QA types (Summary, Generic, Temporal, Short Temporal, Spatial, and Reasoning QA) were defined, with three QA pairs generated per type. For abnormal QA tasks, the other six QA types (Detection, Classification, Temporal, Description, Cause, and Result QA) were applied, but with one QA pair generated per type to emphasize critical semantic cues. Notably, all of the video clips (including normal and abnormal clips) are suitable for the normal QA tasks with their corresponding normal QA pairs, while only abnormal clips are suitable for the abnormal QA tasks with particularly defined abnormal QA pairs. The defined QA categories are illustrated in Figure 4a, and the example questions of each QA task are provided in Figure 4b. The detailed QA pairs generation examples and prompts are shown in Appendix B.

## 3.3 DATA STATISTICS

## 3.4 OVERALL DATA STATISTICS

Table 1 presents statistical information for the SurveillanceVQA-589K (containing MSAD, MEVA, NWPU, and UCA), including the number of videos, total duration, textual annotations, segmented clips (categorized as normal and abnormal), and the total number of QA pairs. More statistical details are presented in Appendix C. Other considerations, such as data quality analysis and limitations, are shown in Appendix G.

Table 1: Overall data statistics of SurveillanceVQA-589K.

| Dataset | Number of Videos | Total Video Duration | Number of Text Annotations | Number of Segments | | QA Pairs |
|---|---|---|---|---|---|---|
| | | | | Normal | Abnormal | |
| MSAD | 201 | 4.23h | 1783 | 1417 | 366 | 34290 |
| MEVA | 720 | 16.76h | 2057 | 2044 | 13 | 37104 |
| NWPU | 255 | 16.29h | 4166 | 4121 | 45 | 75258 |
| UCA | 1854 | 121.9h | 23542 | 20380 | 3162 | 442728 |
| **Total** | **3030** | **159.18h** | **31548** | **27962** | **3586** | **589380** |

### 3.4.1 DISTRIBUTION OF EVENT DURATIONS

Figure 5 illustrates the distribution of event durations on the training/test sets in this dataset. The majority of events are concentrated between 5 and 20 seconds, with a particularly high concentration in the 5-10 seconds and 10-15 seconds intervals. Notably, normal events in the training set dominate the distribution, making up the largest proportion of the dataset.

### 3.4.2 COMPARISONS WITH EXISTING DATASETS

These comparisons, as shown in Table 2, reveal a consistent trend: early video QA datasets are limited in scale, typically containing 200 to 900 videos and 2,000 to 4,000 QA pairs, and primarily focus on normal events with fixed question formats. Moreover, most existing datasets rely exclusively on either human annotations or LLM-generated content, with few employing a hybrid approach that leverages both. UCVL (Chen et al., 2025) serves as an intermediate example by exploring abnormal event QA, yet it still falls short in terms of scale, diversity, and task coverage compared to our SurveillanceVQA-589K dataset.

Table 2: Overall comparisons with existing datasets.

| Aspect | MVBench (Li et al., 2024c) | Video-MME (Fu et al., 2024) | MMB-Video (Fang et al., 2024) | UCVL (Chen et al., 2025) | SurveillanceVQA-589K |
|---|---|---|---|---|---|
| Videos | 200 | 900 | 600 | 1699 | 3030 |
| QA Pairs | 4000 | 2700 | 2000 | 16990 | 589380 |
| Content | Normal | Normal | Normal | Anomaly | Normal&Anomaly |
| QA Forms | MCQ | MCQ | Open-ended | MCQ&Open-ended | Open-ended |
| Generation | LLM | Human | Human | LLM | LLM&Human |
| Evaluation | Matching | Matching | LLM | Matching&LLM | LLM |

# 4 EXPERIMENTS ON SURVEILLANCEVQA-589K

We primarily benchmark eight locally deployed LVLMs and offer suggestions for enhancing their performance. The remaining four API-called LVLMs serve as reference baselines. In our evaluation, we adopt the four key dimensions proposed in the evaluation framework of VideoGPT+ (Maaz et al., 2024). These four key dimensions are Contextual Integration (CI), Detail Orientation (DO) , Contextual Understanding (CU), and Temporal Understanding (TU). We provide the main experimental results and analysis in this section. More detailed experimental settings, results, and explanations have been given in the Appendix D and Appendix E.

## 4.1 RESULTS ON SURVEILLANCEVQA-589K

### 4.1.1 OVERALL LVLMS PERFORMANCE

Table 3 presents evaluation results of various LVLMs across five key dimensions: CI, DO, CU, TU, with LLaVA-Video-7B achieving the highest overall performance due to its integration of the LLaVA and Qwen2 architectures and the AnyRes technique, enhancing image-to-video reasoning. In contrast, InternVL2.5-2B had the lowest average, with weak CI and DO scores, but shows potential in complex reasoning through its Chain-of-Thought mechanism. LLaVA-OV-Qwen2-0.5B, with only 0.5B parameters, achieved a competitive average score, proving that smaller models can still perform well in video comprehension. Qwen2.5-VL-3B-Instruct excelled in multimodal tasks, balancing computational efficiency with contextual understanding. Lastly, Video-LLaMA3-2B/7B improved video representation quality and processing efficiency with advanced features like Any-resolution Vision Tokenization and Differential Frame Pruner.

Table 3: Model performance averaged on different QA tasks across five evaluation dimensions. [†] represents our finetuned LVLMs.

| Method | CI | DO | CU | TU | Avg |
|---|---|---|---|---|---|
| LLaVA-OV-Qwen2-0.5B | 2.89 | 2.62 | 2.89 | 2.64 | 2.76 |
| InternVL2.5-2B | 1.77 | 1.72 | 1.97 | 1.72 | 1.79 |
| VideoLLaMA3-2B | 2.82 | 2.55 | 2.83 | 2.58 | 2.69 |
| Qwen2.5-VL-3B-Instruct | 2.70 | 2.54 | 2.72 | 2.45 | 2.60 |
| LLaVA-NeXT-Video-7B | 2.78 | 2.62 | 2.80 | 2.50 | 2.68 |
| LLaVA-OV-Qwen2-7B | 3.15 | 2.85 | 3.12 | 2.89 | 3.00 |
| LLaVA-Video-7B | 3.17 | 2.85 | 3.15 | 2.92 | **3.02** |
| VideoLLaMA3-7B | 2.93 | 2.67 | 2.93 | 2.70 | 2.80 |
| Qwen2.5-VL-3B-Instruct[†] | 2.83 | 2.71 | 2.83 | 2.58 | 2.74 |
| LLaVA-Video-7B[†] | 3.27 | 3.01 | 3.24 | 3.03 | **3.14** |

### 4.1.2 LVLMS PERFORMANCE ACROSS NORMAL QA TASKS

Table 4 presents the performance of several LVLMs on a range of QA tasks, including Summary, Generic, Temporal, Short Temporal, Spatial, and Reasoning QA. The evaluation is conducted on both normal clips (blue) and abnormal clips (green), with performance metrics reported for each task. Among the models, LLaVA-OV-Qwen2-7B demonstrates the highest overall performance, indicating robust capabilities in handling spatial and reasoning-based questions. In contrast, InternVL2.5-2B exhibits the lowest performance across most tasks, suggesting limitations in processing video-based QA tasks effectively.

A key trend is that models perform better on normal video clips than abnormal ones across most tasks. However, in the Spatial QA, abnormal videos sometimes score higher due to their specific, visually distinctive events (e.g., a person falling or a fight), making spatial grounding easier. Normal

events involve routine or ambiguous contexts, explaining the higher spatial performance on abnormal videos.

Table 4: Performance of different vision-language models across QA tasks. Blue: normal QA tasks on normal video clips, Green: normal QA tasks on abnormal video clips, Brown: abnormal QA tasks on abnormal video clips. † represents our finetuned LVLMs. Green-highlighted models are API-called LVLMs, only tested on abnormal videos.

| Model | Summary | Generic | Temporal | Short Temporal | Spatial | Reasoning | Detection | Classification | Subject | Description | Cause | Result |
|---|---|---|---|---|---|---|---|---|---|---|---|---|
| LLaVA-OV-Qwen2-0.5B | 2.76/2.43 | 2.78/2.52 | 2.62/2.43 | 2.60/2.45 | 3.10/3.12 | 2.93/2.69 | 2.94 | 2.47 | 2.69 | 2.43 | 1.67 | 1.60 |
| InternVL2.5-2B | 0.56/0.37 | 2.20/1.93 | 1.89/1.68 | 2.08/1.95 | 1.92/1.89 | 2.41/2.26 | 1.88 | 1.12 | 1.46 | 0.55 | 0.64 | 0.74 |
| VideoLLaMA3-2B | 2.49/2.00 | 2.84/2.55 | 2.73/2.49 | 2.61/2.44 | 2.97/2.99 | 2.89/2.66 | 1.87 | 2.03 | 2.46 | 1.89 | 1.37 | 1.18 |
| Qwen2.5-VL-3B-Instruct | 2.20/1.49 | 2.66/2.17 | 2.66/2.26 | 2.69/2.31 | 2.86/2.75 | 3.01/2.70 | 1.85 | 2.19 | 2.24 | 1.74 | 1.32 | 1.13 |
| LLaVA-NeXT-7B | 2.17/1.72 | 2.93/2.60 | 2.56/2.28 | 2.66/2.48 | 2.95/2.95 | 3.06/2.81 | 2.32 | 2.56 | 2.59 | 2.11 | 1.96 | 1.65 |
| LLaVA-OV-Qwen2-7B | 3.10/2.79 | 3.08/2.86 | 2.95/2.81 | 2.79/2.66 | 3.31/3.34 | 3.08/2.89 | 2.53 | 2.60 | 2.68 | 2.54 | 1.46 | 1.55 |
| LLaVA-Video-7B | 3.05/2.85 | 2.99/2.80 | 2.85/2.77 | 2.78/2.71 | 3.47/3.51 | 3.25/3.05 | 1.92 | 2.59 | 2.76 | 2.76 | 1.52 | 1.81 |
| VideoLLaMA3-7B | 2.68/2.29 | 3.01/2.70 | 2.81/2.55 | 2.65/2.45 | 3.06/3.06 | 2.99/2.78 | 2.03 | 1.53 | 2.66 | 2.01 | 1.75 | 1.27 |
| Qwen2.5-VL-3B-Instruct† | 2.62/1.99 | 2.76/2.36 | 2.84/2.48 | 2.93/2.59 | 2.89/2.89 | 3.06/2.85 | 1.58 | 2.13 | 2.56 | 2.06 | 1.77 | 1.22 |
| LLaVA-Video-7B† | 3.01/2.63 | 3.16/2.94 | 3.05/2.85 | 3.07/2.86 | 3.41/3.48 | 3.32/3.15 | 4.46 | 3.26 | 3.27 | 2.70 | 1.79 | 1.93 |
| Baidu ERNIE 4.5 Turbo VL | - | - | - | - | - | - | 3.23 | 2.79 | 2.47 | 2.21 | 1.64 | 1.26 |
| Gemini 2.5 Pro | - | - | - | - | - | - | 4.47 | 3.51 | 2.94 | 2.73 | 1.99 | 2.15 |
| GPT-4o | - | - | - | - | - | - | 3.58 | 3.53 | 2.52 | 2.88 | 2.20 | 2.38 |
| InternVL-3.5 | - | - | - | - | - | - | 3.54 | 3.77 | 3.26 | 2.98 | 2.87 | 2.86 |

### 4.1.3 LVLMs performance across abnormal QA tasks

Table 4 also presents the performance of local-deployed and API-called LVLMs on abnormal QA tasks, evaluated exclusively on abnormal video clips with all values reported in Brown. In these scenarios, local-deployed LVLMs show performance variations based on task type. LLaVA-OV-Qwen2-0.5B models excel in Detection tasks, leading in CI and CU metrics. For Classification tasks, both LLaVA-Video-7B and LLaVA-OV-Qwen2-7B perform well. LLaVA-Video-7B excels in Subject and Description questions, while all models struggle with higher-order reasoning like Cause and Result inference. LLaVA-NeXT-7B performs better in causal reasoning, and LLaVA-Video-7B shows slight advantages in result inference. These trends highlight the challenges in understanding causal relationships in abnormal video scenarios. Larger models (7B) generally outperform smaller ones, though specific specializations vary by task. Abnormal QA scores are significantly lower than normal QA scores, reflecting greater difficulty in reasoning with abnormal videos.

### 4.1.4 API-called LVLMs Performance

To further investigate the capability of API-called LVLMs on anomalous video understanding, we conducted an additional set of experiments on anomalous video QA, as reported in the green-highlighted parts of Table 4. In particular, we test the performance of Gemini 2.5 Pro, GPT-4o, ERNIE 4.5 Turbo VL, and InternVL-3.5. InternVL-3.5 outperforms other API-accessed LVLMs, excelling in multimodal causal reasoning. Gemini 2.5 Pro ranks second, while GPT-4o shows weaker performance, especially in the Result QA task. ERNIE-4.5 demonstrates consistent but moderate performance. Locally deployed models, particularly fine-tuned LLaVA-Video variants, perform strongly, showing the competitiveness of the open-source LLaVA series.

## 4.2 Comprehensive analysis

### 4.2.1 Analysis across different LVLMs and used techniques.

Here, we analyze the open-source LVLMs. Smaller models, such as LLaVA-OV-Qwen2-0.5B, excel in CU and DO but struggle with more complex tasks like TU. These models perform well in simpler scenarios but face limitations in long-duration tracking and anomaly detection. Medium models, like

Qwen2.5-VL-3B-Instruct, provide a good balance between performance and efficiency, particularly in handling temporal sequences and contextual integration. Larger models, such as LLaVA-Video-7B and Video-LLaMA3-7B, excel in temporal reasoning and anomaly detection, benefiting from advanced architectures and video-specific fine-tuning.

The experimental results indicate that fine-tuning yields modest performance gains. Fine-tuning models on specific monitoring video data significantly improves their performance in particular environments, helping them better understand context and identify anomalous events. However, fundamentally improving the model's capability to align visual and textual information remains the key to achieving substantial progress. The strengths of these models can largely be attributed to their design choices. Techniques like Differential Frame Pruning and Progressive Scaling help the models handle long video sequences more efficiently while retaining important temporal details. Techniques like Any-Resolution Vision Tokenization, are capable of processing low-resolution video, suitable for monitoring applications where video quality may vary.

### 4.2.2 ANALYSIS ACROSS QA TYPES AND VIDEO CONTEXTS.

The performance of vision-language models varies significantly across different QA types and video contexts. Normal video QA tasks, such as Summary, Generic, Spatial, and Reasoning QA, generally achieve higher scores across all models. For instance, models like LLaVA-OV-Qwen2-7B and LLaVA-Video-7B consistently score above 3.0 on Spatial and Generic QA, indicating strong capabilities in visual description and object localization in well-structured scenes. In contrast, abnormal video QA tasks, especially Cause and Result QA, remain highly challenging. Across all models, scores for Cause QA fall below 2.0, with InternVL2.5-2B as low as 0.64, and the best-performing model (LLaVA-NeXT-7B) only reaching 1.96. Even the SOTA API-called LVLMs still show low scores for Cause and result QA. This suggests a widespread limitation in causal reasoning, especially under chaotic or low-frequency events like violence or accidents. Moreover, models generally perform better on normal video clips than on abnormal ones, even for the same QA types. This indicates that scene stability significantly influences model comprehension.

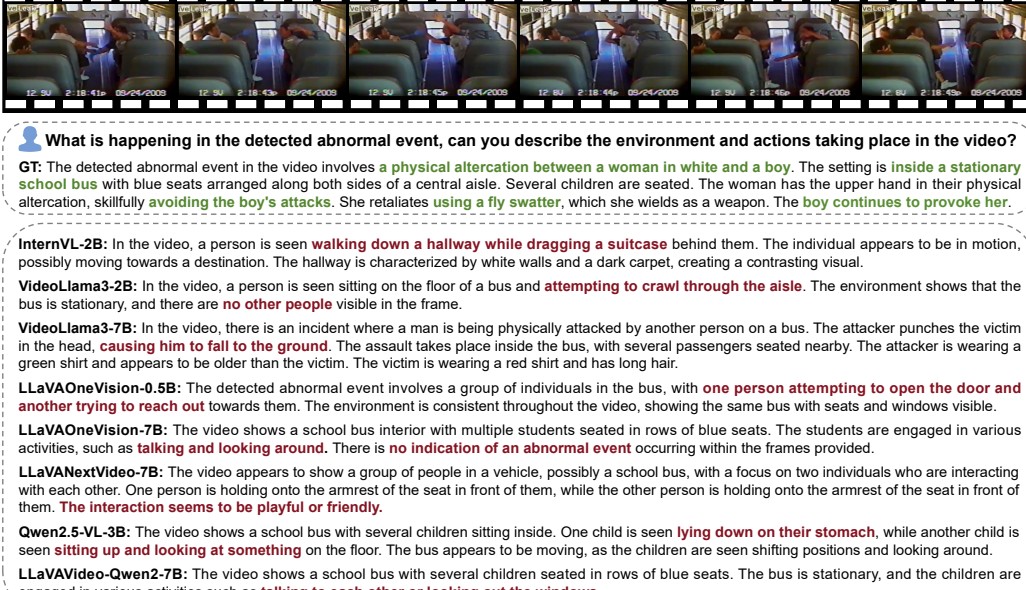

👤 **What is happening in the detected abnormal event, can you describe the environment and actions taking place in the video?**

**GT:** The detected abnormal event in the video involves **a physical altercation between a woman in white and a boy**. The setting is **inside a stationary school bus** with blue seats arranged along both sides of a central aisle. Several children are seated. The woman has the upper hand in their physical altercation, skillfully **avoiding the boy's attacks**. She retaliates **using a fly swatter**, which she wields as a weapon. The **boy continues to provoke her**.

**InternVL-2B:** In the video, a person is seen **walking down a hallway while dragging a suitcase** behind them. The individual appears to be in motion, possibly moving towards a destination. The hallway is characterized by white walls and a dark carpet, creating a contrasting visual.

**VideoLlama3-2B:** In the video, a person is seen sitting on the floor of a bus and **attempting to crawl through the aisle**. The environment shows that the bus is stationary, and there are **no other people** visible in the frame.

**VideoLlama3-7B:** In the video, there is an incident where a man is being physically attacked by another person on a bus. The attacker punches the victim in the head, **causing him to fall to the ground**. The assault takes place inside the bus, with several passengers seated nearby. The attacker is wearing a green shirt and appears to be older than the victim. The victim is wearing a red shirt and has long hair.

**LLaVAOneVision-0.5B:** The detected abnormal event involves a group of individuals in the bus, with **one person attempting to open the door and another trying to reach out** towards them. The environment is consistent throughout the video, showing the same bus with seats and windows visible.

**LLaVAOneVision-7B:** The video shows a school bus interior with multiple students seated in rows of blue seats. The students are engaged in various activities, such as **talking and looking around**. There is **no indication of an abnormal event** occurring within the frames provided.

**LLaVANextVideo-7B:** The video appears to show a group of people in a vehicle, possibly a school bus, with a focus on two individuals who are interacting with each other. One person is holding onto the armrest of the seat in front of them, while the other person is holding onto the armrest of the seat in front of them. **The interaction seems to be playful or friendly.**

**Qwen2.5-VL-3B:** The video shows a school bus with several children sitting inside. One child is seen **lying down on their stomach**, while another child is seen **sitting up and looking at something** on the floor. The bus appears to be moving, as the children are seen shifting positions and looking around.

**LLaVAVideo-Qwen2-7B:** The video shows a school bus with several children seated in rows of blue seats. The bus is stationary, and the children are engaged in various activities such as **talking to each other or looking out the windows.**

Figure 6: Abnormal videos QA Examples. All of the local-deployed LVLMs provide wrong answers.

### 4.2.3 ANALYSIS OF FINETUNED LVLMs

The experimental results demonstrate the impact of fine-tuning on LVLMs across various QA tasks. After LoRA fine-tuning, Qwen2.5-VL-3B-Instruct[†] and LLaVA-Video-7B[†] achieved better average

scores. The fine-tuned 7B model outperforms all API-called models in the anomaly detection task, and it also highlights the significance of our dataset research in improving model performance.

However, fine-tuning results in performance degradation in certain QA types. This indicates that during fine-tuning, models may experience catastrophic forgetting of general knowledge acquired during pre-training, leading to over-adaptation to the fine-tuning dataset. Furthermore, in visual cognition tasks, while fine-tuning can enhance performance on target tasks, it does not necessarily yield human-like robust generalization, particularly in complex QA categories such as causal reasoning or anomaly classification. This highlights a limitation of current fine-tuning approaches.

### 4.2.4 ANALYSIS OF FAILED CASES

In Figure 6, the model responses to the abnormal event detection in the video demonstrate several errors and misinterpretations of the scene. Notably, multiple models provide descriptions of events that deviate from the actual content of the video. These errors stem from misinterpreting both the environment and the individuals involved in the scene. Most models struggle with distinguishing the abnormal event from general bus activity, demonstrating the challenge of correctly identifying nuanced interactions in complex video scenes. This highlights the research challenges of LVLMs for understanding our proposed SurveillanceVQA-589K dataset. We have also noticed the poor performance of LVLMs on causal reasoning QA tasks. Therefore, we present more relevant failed cases and analysis in Appendix F.

### 4.3 SUGGESTIONS FOR FUTURE RESEARCH

In addition to the findings presented in this work, we recognize that advancing causal reasoning and domain-adaptive pretraining strategies constitutes an important direction for future research. On the one hand, causal reasoning can be further enhanced through structured temporal event modeling and step-by-step inference mechanisms, enabling models to explicitly capture event segmentation, temporal dependencies, and causal transitions in surveillance video. Such approaches have the potential to yield more coherent and focused causal explanations while mitigating redundant multi-hypothesis outputs. On the other hand, domain-adaptive pretraining offers a promising pathway to bridge the gap between generic video understanding and the unique challenges of surveillance contexts. In particular, refining visual encoding networks to emphasize salient cues, improving video and language interaction tailored to surveillance, and adopting coarse-to-fine pretraining pipelines from captioning to question answering are expected to strengthen the adaptability and robustness of future systems. These directions highlight promising avenues for building more reliable and context-aware surveillance video understanding models.

## 5 CONCLUSION

In this study, we introduce SurveillanceVQA-589K, the largest open-ended video QA benchmark tailored specifically to real-world surveillance scenarios. The dataset contains 589,380 QA pairs spanning 12 cognitively diverse task types across both normal and abnormal surveillance video contexts. We propose a hybrid annotation pipeline that combines human-aligned captions with LVLM-assisted QA generation, enabling high-quality, scalable annotation. We benchmark eight local-deployed LVLMs ( ranging from 0.5B to 7B parameters) and four API-called LVLMs. Our experiments reveal that while these models demonstrate promising performance on general understanding tasks, they struggle significantly with complex semantic reasoning, particularly in anomaly-specific tasks such as causal inference and result prediction, indicating a clear performance bottleneck in high-level temporal and logical reasoning. We also examine the impact of fine-tuning through LoRA on a 3B/7B model. While fine-tuning yields moderate gains on general tasks, it provides limited improvement in structured anomaly detection and classification, highlighting that current parameter-efficient tuning approaches are insufficient for enabling complex and specific-domain tasks. Overall, this work provides a comprehensive testbed for evaluating multimodal models in realistic surveillance settings. Our findings underscore the pressing need for LVLMs to develop stronger causal reasoning, temporal modeling, and structured response generation capabilities. Advancing causal reasoning and domain-adaptive pretraining strategies constitutes an important direction for future research.

ETHICAL STATEMENT

In this study, we strictly adhere to the ethical guidelines published by the ICLR that address key ethical considerations throughout our data construction, model evaluation, and sharing processes. The video datasets used (such as MEVA, MSAD, NWPU, and UCF) are all publicly available and intended for research purposes, with no personally identifiable information involved, ensuring the legality and compliance of data sources. To mitigate potential biases, we incorporated diverse video scenarios and carefully crafted prompts during the QA generation and evaluation stages, aiming to reduce cultural or contextual bias inherent in language models or the original datasets. Moreover, we explicitly oppose any unauthorized use of our dataset or methods (e.g., for abusive surveillance or discriminatory profiling), and we deliberately avoided including high-risk content such as facial recognition. During data annotation, we employed a collaborative pipeline combining human annotators and large language models (e.g., Qwen-Max), with humans responsible for event segmentation and verification, and fair compensation provided in accordance with local wage standards. We plan to release our dataset and code under a research-friendly license with clear ethical usage guidelines, promoting responsible practices in multimodal research.

REPRODUCIBILITY STATEMENT

We are committed to ensuring the reproducibility of our work by the broader research community. The video datasets used in this research, including MEVA, MSAD, NWPU, and UCF, are all publicly available and intended exclusively for research purposes. We have made our dataset available under a research-friendly license, which can be accessed at `https://anonymous.4open.science/ r/SurveillanceVQA-589K`. Additionally, the large vision-language models (LVLMs) tested in our study are either open-sourced or accessible via APIs, allowing for transparency and easy replication of our experiments. By providing these resources and ensuring the availability of all necessary components, we aim to facilitate the independent validation and further exploration of our findings within the research community.

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

# Appendix

## A  VIDEO ANNOTATION GENERATION

### A.1  HUMAN ANNOTATION GUIDELINES AND QUALITY ASSURANCE

To ensure annotation accuracy and consistency, we provided annotators with clear guidelines and comprehensive training, following the protocol established in UCA (Yuan et al., 2024b). The annotation was carried out by a team of technically proficient annotators who were fairly compensated according to local wage standards. The process was supervised by AI researchers who regularly reviewed and validated the outputs to ensure both quality and ethical compliance. The entire annotation phase spanned approximately one month and resulted in a high-quality corpus. We then integrated this newly annotated data with existing annotations from UCA, completing the manual annotation collection with a total of 31,548 sentence-level annotations accompanied by precise timestamps.

The annotation guideline design includes:

- Fine-grained annotation principle
- Rich sentence descriptions
- Region of Interest (ROI) descriptions
- Handling intense visual changes
- Complex environment descriptions.

The quality assurance measures include:

- Periodic validation checks conducted by the review team every 100 instances
- Cross-annotator consistency monitoring and resolution of discrepancies

We believe that this rigorous approach to annotation, with the combination of clear guidelines, comprehensive training, and robust quality assurance procedures, will help users better assess the quality and reliability of the human-labeled portion of our dataset. By providing transparent insights into our annotation process, we aim to foster greater trust and confidence in the dataset, facilitating its use in various research and development applications.

### A.2  VARIOUS ANNOTATION DATA EXAMPLES

The examples of video sources in our SurveillanceVQA-589K are shown in Figure 7, including UCF-Crime (Sultani et al., 2018), MEVA (Corona et al., 2021), NWPU Campus dataset (Cao & Others, 2023), and MSAD (Zhu et al., 2024). The annotation text data examples including manually annotations, detailed description by LLaVA-Next, and intergrated annotations by Qwen-turbo, are shown in the following lists.

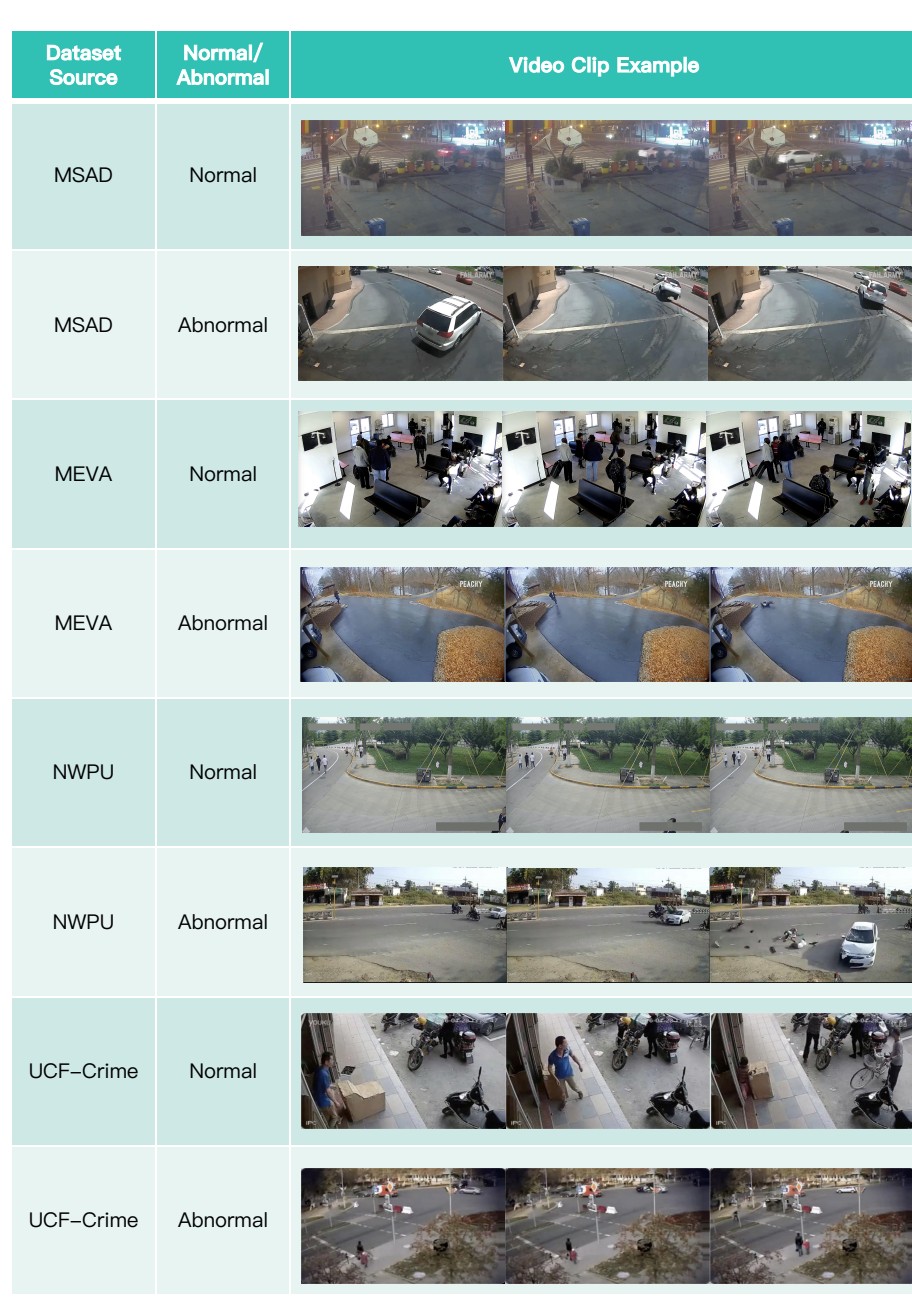

Figure 7: Video clip examples from four video datasets.

```
1  Example of manually annotations:
2
3  "2018-03-15_10_131": {
4      "timestamps": [
5          [3.2, 7.7],
6          [50.0, 54.0],
7          [114.0, 118.0],
8          [148.0, 156.0],
9          [188.0, 194.0]
10     ],
11     "sentences": [
12         "A man in a hat goes down to the underground passage.",
13         "A cuckold man came up from the tunnel crossing.",
14         "A woman with long hair carried a bag down the underground
               passage.",
15         "A blue dress white hair man net underground passage down",
16         "Two men came up from the underpass"
17     ]
18 }
```

```
1  Example of Detailed Description by LLaVA-Next:
2  {
3      "video_name": "2018-03-15_10_131_5.mp4",
4      "duration": 6.0,
5      "description": "The video opens with a view of a concrete
               staircase leading down to a dark tunnel entrance, flanked by
               metal railings on both sides. The ground surrounding the
               stairs is covered in grass. Two individuals are present at the
               top of the stairs; one is wearing a green jacket and a beanie
               , while the other is dressed in black clothing and a white hat
               . The person in the green jacket begins to descend the stairs
               first, followed by the individual in black. As they move down,
               the person in the green jacket pauses midway, looking back
               towards the camera, while the person in black continues their
               descent. The video concludes with both individuals still
               descending the stairs, with the person in the green jacket
               still pausing and looking back."
6  }
```

```
1  Example of the intergrated annotations by Qwen-turbo:
2  "2018-03-15_10_131": {
3      "timestamps": [
4          [
5              3.2,
6              7.7
7          ],
8          [
9              50.0,
10             54.0
11         ],
12         [
13             114.0,
14             118.0
```

```
15              ],
16              [
17                      148.0,
18                      156.0
19              ],
20              [
21                      188.0,
22                      194.0
23              ]
24          ],
25          "sentences": [
26              "A man wearing dark clothing and a beanie hat walks down
                    a set of concrete stairs enclosed by metal railings on
                     either side. The stairs feature red handrails, and
                    the man descends them steadily. As he moves downward
                    ......",
27              "The video begins with a view of a concrete staircase
                    descending into a dark tunnel. Metal railings line
                    both sides of the staircase, and the area at the top
                    of the stairs is overgrown with grass......",
28              "The video begins with a view of a concrete staircase
                    leading down to a dark tunnel entrance.........",
29              "The video begins with a view of a concrete staircase
                    leading down to a dark tunnel entrance. Metal railings
                     flank both sides of the staircase......",
30              "......The video ends with both individuals still moving
                    down the stairs, with the person in the green jacket
                    continuing to pause and look back."
31          ]
32      }
```

## B   AUTOMATIC QA GENERATION

For all the videos(including normal/abnormal clips), the generated JSON file includes the following normal QA categories:

- summary_qa_pairs: Cover the full narrative of the video, summarizing the entire sequence of events.

- generic_qa_pairs: Focus on essential visual and behavioral information, including appearance, actions, trajectories, and inferred intentions.

- temporal_qa_pairs: Address the order and timing of events, using general time references (e.g., beginning, middle, end).

- spatial_qa_pairs: Explore spatial details such as clothing colors, physical positions, and scene layout.

- reasoning_qa_pairs: Emphasize causal and inferential questions related to actions, locations, and motivations ("what", "where", "why").

- short_temporal_qa_pairs: Provide concise questions about specific moments or transitions within the video.

The corresponding JSON file for this abnormal case includes the following abnormal QA categories and content:

| Type | Task Description | Key Instructions | Example Question | Example Answer | Source |
|---|---|---|---|---|---|
| Summary QA | Generate questions to extract a detailed description of the entire video content. | Generate three questions targeting the full sequence, with answers integrating all details. | Can you describe the entire video in detail from start to finish? | The video begins with a long, well–lit hallway featuring light–colored walls and a shiny floor. A woman in black, enters the frame from the right side... | MEVA_QA/2018–05–16_14_187_5.json |
| Generic QA | Generate questions focusing on significant aspects like appearance and motion. | Generate three questions on different aspects (appearance, motion, reasoning), with detailed answers. | Describe the appearance and activities of all individuals in the video. | In the video, there are four individuals. On the lower side, a woman in black walks upward through the lower side.... | MEVA_QA/2018–05–16_14_187_5.json |
| Temporal QA | Generate questions focusing on the sequence and timing of events. | Generate three questions using time references (beginning, middle, end), with answers based on the caption. | What actions occur after the initial setup, and how do they progress towards the climax of the video? | After the initial setup, where the man is seen walking away and the woman in black with the white bag passes him... | MEVA_QA/2018–05–16_14_187_5.json |
| Short Temporal QA | Generate concise questions focusing on specific temporal events in the video. | Generate three questions on temporal aspects using approximate time references, with answers based on the caption. | When does the man in dark clothing start walking back towards the camera? | The man walking back towards the camera, which happens when the woman in black, carrying a white bag, approaches him. This occurs in the middle of the video... | MEVA_QA/2018–05–16_14_187_5.json |
| Spatial QA | Generate questions focusing on spatial details like colors and outfits. | Generate three questions on spatial aspects (colors, attire, location), with detailed answers. | Describe the setting of the video and the objects in the scene. | The video is set in a long, well–lit hallway with light–colored walls and a shiny floor. | MEVA_QA/2018–05–16_14_187_5.json |
| Reasoning QA | Generate questions focusing on actions, objects, and reasoning behind events. | Generate three questions on actions, objects, and reasoning, with concise answers including context. | What is the man in black doing at the end of the video? | The man in black is walking further down the hallway and eventually moves out of sight. | MEVA_QA/2018–05–16_14_187_5.json |

Figure 8: Six normal QA examples and designed prompts.

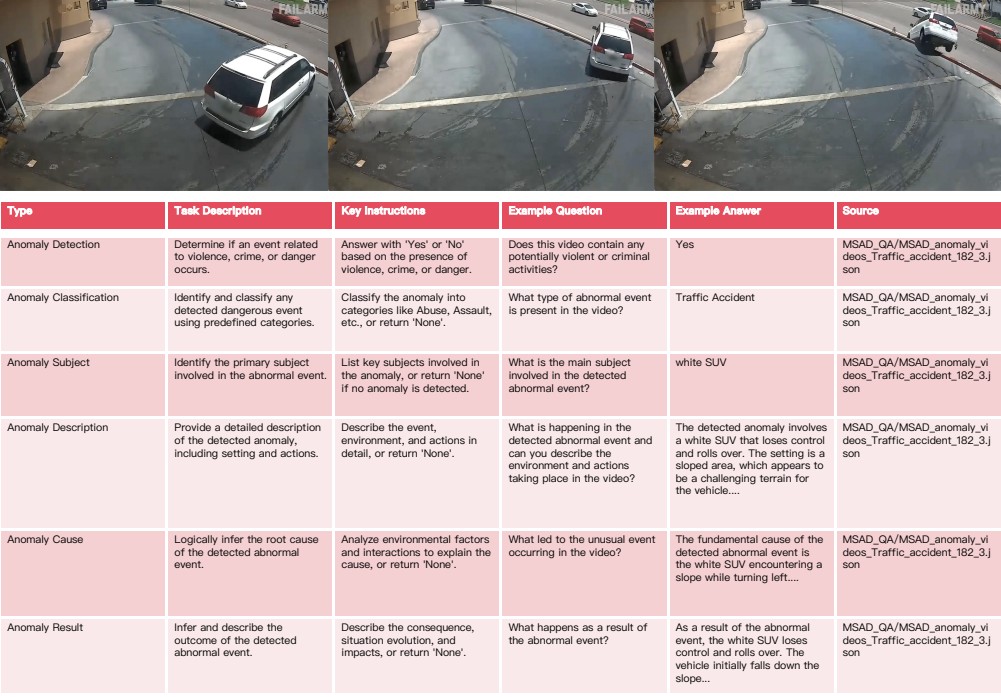

| Type | Task Description | Key Instructions | Example Question | Example Answer | Source |
|---|---|---|---|---|---|
| Anomaly Detection | Determine if an event related to violence, crime, or danger occurs. | Answer with 'Yes' or 'No' based on the presence of violence, crime, or danger. | Does this video contain any potentially violent or criminal activities? | Yes | MSAD_QA/MSAD_anomaly_videos_Traffic_accident_182_3.json |
| Anomaly Classification | Identify and classify any detected dangerous event using predefined categories. | Classify the anomaly into categories like Abuse, Assault, etc., or return 'None'. | What type of abnormal event is present in the video? | Traffic Accident | MSAD_QA/MSAD_anomaly_videos_Traffic_accident_182_3.json |
| Anomaly Subject | Identify the primary subject involved in the abnormal event. | List key subjects involved in the anomaly, or return 'None' if no anomaly is detected. | What is the main subject involved in the detected abnormal event? | white SUV | MSAD_QA/MSAD_anomaly_videos_Traffic_accident_182_3.json |
| Anomaly Description | Provide a detailed description of the detected anomaly, including setting and actions. | Describe the event, environment, and actions in detail, or return 'None'. | What is happening in the detected abnormal event and can you describe the environment and actions taking place in the video? | The detected anomaly involves a white SUV that loses control and rolls over. The setting is a sloped area, which appears to be a challenging terrain for the vehicle.... | MSAD_QA/MSAD_anomaly_videos_Traffic_accident_182_3.json |
| Anomaly Cause | Logically infer the root cause of the detected abnormal event. | Analyze environmental factors and interactions to explain the cause, or return 'None'. | What led to the unusual event occurring in the video? | The fundamental cause of the detected abnormal event is the white SUV encountering a slope while turning left.... | MSAD_QA/MSAD_anomaly_videos_Traffic_accident_182_3.json |
| Anomaly Result | Infer and describe the outcome of the detected abnormal event. | Describe the consequence, situation evolution, and impacts, or return 'None'. | What happens as a result of the abnormal event? | As a result of the abnormal event, the white SUV loses control and rolls over. The vehicle initially falls down the slope... | MSAD_QA/MSAD_anomaly_videos_Traffic_accident_182_3.json |

Figure 9: Six abnormal QA examples and designed prompts.

- get_anomaly_detection: Determines whether the video contains violent, criminal, or dangerous events.
- get_anomaly_classification_prompt: Classifies the anomaly type—in this case, as a shooting.
- get_anomaly_subject: Identifies the key individuals involved in the anomaly—two men on a motorcycle (initiators with firearms) and a woman who returns fire.
- get_anomaly_description: Provides a detailed account of the shooting (focused between 35.1–41.0 seconds), describing the environment (street), appearances (black clothing), and actions (threatening, shooting, fleeing).
- get_anomaly_cause: Infers the likely cause of the anomaly—here, the armed threat initiated by the motorcyclists.
- get_anomaly_result: Analyzes the consequence or outcome of the anomalous event.

This streamlined QA structure for abnormal segments allows the dataset to efficiently highlight critical details necessary for real-time anomaly detection and situational understanding, contrasting with the more comprehensive QA setup used for normal videos.

Here are the QA generation examples:

In Figure 8, for normal/abnormal video clips, six normal QA categories are defined, with three QA pairs in each category to ensure a detailed and rich representation of key elements. Take the MEVA video "2018-05-16_14_187_5.mp4" as an example. This video records the scene of a woman wearing black clothes and carrying a white backpack passing by in a corridor. During this process, some other pedestrians walked along the corridor.

In Figure 9, for abnormal video clips, six abnormal QA categories are defined, with one QA pair for each category to ensure a concise and focused representation of key abnormal elements. Take the MSAD video "MSAD_anomaly_videos_Traffic_accident_182_3.mp4" as an example. This video describes a traffic accident that occurred on the road, recording the process in which a vehicle (a white SUV) lost control and overturned.

## C  MORE DATA STATISTICS

### C.1  CATEGORIZATION STATISTICS

In terms of dataset partitioning, we split the dataset at the clip level rather than by entire videos, using an 8/2 ratio for the training and testing sets. In terms of dataset partitioning, we perform splitting at the clip level with an 8/2 ratio for the training and testing sets. Importantly, to prevent data leakage, all clips originating from the same original video are strictly assigned to the same split, ensuring that no video contributes segments to both the training and test sets.

| Category | MEVA | MSAD | NWPU | UCA |
|---|---|---|---|---|
| **normal** | 2044 | 1417 | 4121 | 20384 |
| **Abuse** | 0 | 1 | 0 | 138 |
| **Arrest** | 0 | 2 | 0 | 143 |
| **Assault** | 0 | 38 | 4 | 451 |
| **Burglary** | 0 | 7 | 4 | 204 |
| **Explosion** | 0 | 15 | 0 | 73 |
| **Fighting** | 1 | 34 | 6 | 400 |
| **Fire** | 0 | 52 | 0 | 217 |
| **Object Falling** | 1 | 55 | 1 | 101 |
| **People Falling** | 3 | 109 | 9 | 570 |
| **Pursuit** | 0 | 1 | 1 | 19 |
| **Robbery** | 1 | 63 | 7 | 575 |
| **Shooting** | 0 | 15 | 0 | 56 |
| **Shoplifting** | 0 | 0 | 0 | 26 |
| **Stealing** | 2 | 5 | 13 | 250 |
| **Traffic Accident** | 4 | 52 | 2 | 441 |
| **Threat** | 0 | 1 | 0 | 7 |
| **Vandalism** | 2 | 25 | 2 | 255 |
| **Water Incident** | 0 | 6 | 0 | 6 |

Table 5: Event Category statistics on video clips.

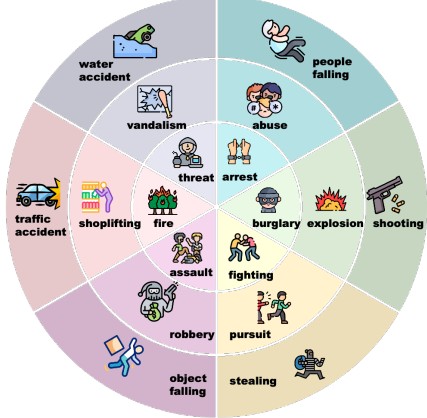

Figure 10: Different types of abnormal video clips.

Due to the difficulty of annotating every second of footage, the total duration of the training and testing sets is slightly shorter than that of the original raw video collection. Table 6 presents the distribution of QA pairs for normal and abnormal events across the training and testing sets. This table reports the number of QA pairs generated for both event types and illustrates how their distribution varies between subsets for each dataset. Here, the normal category refers to non-anomalous clips, while the abnormal category corresponds to anomalous clips, which span 18 distinct abnormal event classes as detailed in Figure 10.

Regarding video event categorization, as shown in Table 5 in addition to the Normal category representing non-anomalous cases, we establish a classification system comprising 18 distinct abnormal categories for the Abnormal class. These include: Abuse, Arrest, Assault, Burglary, Explosion, Fighting, Fire, Object Falling, People Falling, Pursuit, Robbery, Shooting, Shoplifting, Stealing, Traffic Accident, Threat, Vandalism, and Water Incident—covering a wide range of incidents from violent behaviors (e.g., Assault, Fighting) to environmental hazards (e.g., Fire, Water Incident).

Notably, we have observed that some abnormal categories generated by Qwen-Max exhibit redundancy or lack general applicability. To improve the accuracy of statistical analysis and standardize the taxonomy, we perform category consolidation and refinement. For instance, "Chasing" and "Chase" are unified under Pursuit due to semantic equivalence, while overly broad categories such as "Emergency Situation" and "Weapon Present" are excluded from statistical summaries due to their ambiguity in defining concrete abnormal behaviors. It is important to emphasize that this refinement is applied only during the statistical analysis phase, and the original Qwen-Max outputs remain unaltered.

Table 6: Distribution of QA pairs for normal and abnormal events across the training and testing sets

|  | MEVA | | MSAD | | NWPU | | UCA | |
|---|---|---|---|---|---|---|---|---|
|  | Abnormal | Normal | Abnormal | Normal | Abnormal | Normal | Abnormal | Normal |
| Test | 72 | 7362 | 1776 | 5112 | 240 | 14850 | 15192 | 73368 |
| Train | 240 | 29430 | 7008 | 20394 | 840 | 59328 | 60696 | 293472 |

# D  DETAILED EXPERIMENT SETTINGS

## D.1  EVALUATION DESIGN

In our evaluation, we adopt the four key dimensions proposed in the evaluation framework of VideoGPT+ (Maaz et al., 2024). These four key dimensions are listed as follows. Contextual Integration (CI) measures whether the answer accurately reflects the factual content of the video, avoiding errors or misinterpretations. Detail Orientation (DO) assesses the inclusion of specific and complete key elements. Contextual Understanding (CU) evaluates the alignment of the answer with the overall narrative and emotional tone of the video. Temporal Understanding (TU) focuses on the correctness of event sequences and time-related logic. Unlike the original framework, we do not include the consistency evaluation dimension, since our task focuses on generating diverse and representative question–answer pairs rather than measuring consistency across highly similar or repetitive QA pairs. Each dimension is rated on a 0–5 integer scale, with 5 indicating full accuracy and relevance, and 0 indicating a completely incorrect response. To calculate Average Score (Avg), individual scores are normalized by multiplying each by 0.25 and summing the results. This scoring scheme enables fine-grained, quantitative evaluation of model performance across multiple facets of video understanding. During the evaluation phase, we further employ an LLM-based strategy using the GLM-4-Flash API (GLM et al., 2024), which allows us to capture both semantic consistency and the interpretive and reasoning capabilities of advanced language models.

## D.2  BASELINES AND SETTINGS

We evaluate 8 open-source video-language models by locally deploying them on our devices, including the VideoLLaMA3 (Zhang et al., 2025), InternVL2.5 (Chen et al., 2024c), LLaVA-OV-Qwen2 (Li et al., 2024a)), LLaVA-Video-Qwen2 (Zhang et al., 2024c), LLaVA-NeXT-Video (Zhang et al.,

2024a), and Qwen2.5-VL-Instruct series (Bai et al., 2025), with parameter sizes ranging from 0.5B to 7B. Each model performs inference on one question at a time to prevent information leakage between questions. More detailed settings are shown in Appendix D. In addition, we further evaluated other API-accessed LVLMs on the abnormal video QA task, including Google DeepMind's Gemini 2.5 Pro Google (2025) , OpenAI's GPT-4o OpenAI (2024) , Baidu's ERNIE 4.5 Turbo VL Baidu (2025) , and the newest model InternVL-3.5 Wang et al. (2025) from Shanghai AI Lab.

In the evaluation experiment, we adopted the default hyperparameter configurations of each model to ensure a fair comparison. The only modification is the adjustment of the input video frame rate to adapt to the GPU memory limit. Specifically, LLaVA-OneVision-0.5B/7B, LLaVA-NeXT-Video-7B and LLaVA-Video-7B use 32 frames, while InternVL2.5-2B uses 24 frames. VideoLLaMA3-2B/7B samples up to 512 frames at 1 fps, while Qwen2.5-VL-3B-Instruct adopts its default dynamic frame sampling strategy. All benchmark tests were conducted on NVIDIA RTX 4090 GPUs.

In terms of fine-tuning, we first selected Qwen2.5-VL-Instruct-3B as the benchmark model and conducted LoRA fine-tuning on a single NVIDIA RTX 4090 GPU. During the training process, we only fine-tune the language model components and the vision-language fusion module, while keeping the visual encoder frozen to reduce training costs and stabilize performance. The training uses `bfloat16` precision to enhance the efficiency of the video memory and imposes resolution constraints on the multimodal image input (maximum number of pixels 50,176, minimum number of pixels 784) to avoid GPU memory overload caused by high-resolution input. The model was trained for one epoch, with the hyperparameters set as:

$$\text{batch\_size} = 1, \quad \text{gradient\_accumulation\_steps} = 8, \quad \text{learning\_rate} = 2 \times 10^{-7}.$$

Furthermore, we fine-tuned the LLaVA-Video-7B model for LoRA on a single NVIDIA RTX 5090 GPU, training only the linear layer, and the training lasted for a total of 3 epochs. Its hyperparameters are set as follows:

$$\text{batch\_size} = 4, \quad \text{gradient\_accumulation\_steps} = 1, \quad \text{learning\_rate} = 1 \times 10^{-5}.$$

### D.3 DESCRIPTIONS OF LOCAL-DEPLOYED LVLMS

Table 7 shows the characteristics of these evaluated local-deployed LVLMs.

Table 7: Overview of Evaluated Open-Source Video Models

| Model Name | Key Features |
|---|---|
| Video-LLaMA3-2B/7B (Zhang et al., 2025) | Uses any-resolution vision tokenization and differential frame pruner to reduce information loss and computation cost. Trained on high-quality video data. |
| InternVL2.5-2B (Chen et al., 2024c) | Based on the ViT-MLP-LLM framework. Incorporates dynamic high-resolution representations, Progressive scaling Strategy, and Chain-of-Thought (CoT) reasoning. |
| LLaVA-OV-Qwen2-0.5B/7B (Li et al., 2024a)) | Multimodal model capable of understanding single images, multiple images, and videos. Supports cross-modal transfer learning. |
| LLaVA-Video-7B (Zhang et al., 2024c) | Trained only on text-image data with AnyRes technology. Fine-tuned on LLaVA-Video-178K for enhanced video instruction understanding. |
| Qwen2.5-VL-3B-Instruct (Bai et al., 2025) | Strong instruction-following capabilities across text, image, and video. Improved cross-modal alignment and QA generation. |
| LLaVA-NeXT-Video-7B (Zhang et al., 2024a) | A strong zero-shot video understanding Model |

## E MORE EXPERIMENT RESULTS

### E.1 LVLMS PERFORMANCE RANKING

Figure 11 shows an overall performance ranking of Local-deployed LVLMs across different QA tasks.

| Rank | Model | Avg |
|------|-------|-----|
| 1 | LLavA-OV-Qwen2-7B | 3.10 |
| 2 | LLaVA-Video-7B-Qwen2 | 3.05 |
| 3 | LLavA-OV-Qwen2-0.5B | 2.76 |
| 4 | VideoLLaMA3-7B | 2.68 |
| 5 | VideoLLaMA3-2B | 2.49 |
| 6 | Qwen2.5-VL-3B | 2.20 |
| 7 | LLaVA-Next-7B | 2.17 |
| 8 | InternVL2.5-2B | 0.56 |

Table 1: Summary QA

| Rank | Model | Avg |
|------|-------|-----|
| 1 | LLavA-OV-Qwen2-7B | 3.08 |
| 2 | VideoLLaMA3-7B | 3.01 |
| 3 | LLaVA-Video-7B-Qwen2 | 2.99 |
| 4 | LLaVA-Next-7B | 2.93 |
| 5 | VideoLLaMA3-2B | 2.84 |
| 6 | LLavA-OV-Qwen2-0.5B | 2.78 |
| 7 | Qwen2.5-VL-3B | 2.66 |
| 8 | InternVL2.5-2B | 2.20 |

Table 2: Generic QA

| Rank | Model | Avg |
|------|-------|-----|
| 1 | LLavA-OV-Qwen2-7B | 2.95 |
| 2 | LLaVA-Video-7B-Qwen2 | 2.85 |
| 3 | VideoLLaMA3-7B | 2.81 |
| 4 | VideoLLaMA3-2B | 2.73 |
| 5 | Qwen2.5-VL-3B | 2.66 |
| 6 | LLavA-OV-Qwen2-0.5B | 2.62 |
| 7 | LLaVA-Next-7B | 2.56 |
| 8 | InternVL2.5-2B | 1.89 |

Table 3: Temporal QA

| Rank | Model | Acc |
|------|-------|-----|
| 1 | LLavA-OV-Qwen2-7B | 2.79 |
| 2 | LLaVA-Video-7B-Qwen2 | 2.78 |
| 3 | Qwen2.5-VL-3B | 2.69 |
| 4 | LLaVA-Next-7B | 2.66 |
| 5 | VideoLLaMA3-7B | 2.65 |
| 6 | VideoLLaMA3-2B | 2.61 |
| 7 | LLavA-OV-Qwen2-0.5B | 2.60 |
| 8 | InternVL2.5-2B | 2.08 |

Table 4: Short Temporal

| Rank | Model | Acc |
|------|-------|-----|
| 1 | LLaVA-Video-7B-Qwen2 | 3.47 |
| 2 | LLavA-OV-Qwen2-7B | 3.31 |
| 3 | LLavA-OV-Qwen2-0.5B | 3.10 |
| 4 | VideoLLaMA3-7B | 3.06 |
| 5 | VideoLLaMA3-2B | 2.97 |
| 6 | LLaVA-Next-7B | 2.95 |
| 7 | Qwen2.5-VL-3B | 2.86 |
| 8 | InternVL2.5-2B | 1.92 |

Table 5: Spatial QA

| Rank | Model | Acc |
|------|-------|-----|
| 1 | LLaVA-Video-7B-Qwen2 | 3.25 |
| 2 | LLavA-OV-Qwen2-7B | 3.08 |
| 3 | LLaVA-Next-7B | 3.06 |
| 4 | Qwen2.5-VL-3B | 3.01 |
| 5 | VideoLLaMA3-7B | 2.99 |
| 6 | LLavA-OV-Qwen2-0.5B | 2.93 |
| 7 | VideoLLaMA3-2B | 2.89 |
| 8 | InternVL2.5-2B | 2.41 |

Table 6: Reasoning QA

| Rank | Model | Avg |
|------|-------|-----|
| 1 | LLaVA-Video-7B-Qwen2 | 2.85 |
| 2 | LLavA-OV-Qwen2-7B | 2.79 |
| 3 | LLavA-OV-Qwen2-0.5B | 2.43 |
| 4 | VideoLLaMA3-7B | 2.29 |
| 5 | VideoLLaMA3-2B | 2.00 |
| 6 | LLaVA-Next-7B | 1.72 |
| 7 | Qwen2.5-VL-3B | 1.49 |
| 8 | InternVL2.5-2B | 0.37 |

Table 7: Summary QA

| Rank | Model | Avg |
|------|-------|-----|
| 1 | LLavA-OV-Qwen2-7B | 2.86 |
| 2 | LLaVA-Video-7B-Qwen2 | 2.80 |
| 3 | VideoLLaMA3-7B | 2.70 |
| 4 | LLaVA-Next-7B | 2.60 |
| 5 | VideoLLaMA3-2B | 2.55 |
| 6 | LLavA-OV-Qwen2-0.5B | 2.52 |
| 7 | Qwen2.5-VL-3B | 2.17 |
| 8 | InternVL2.5-2B | 1.93 |

Table 8: Generic QA

| Rank | Model | Avg |
|------|-------|-----|
| 1 | LLavA-OV-Qwen2-7B | 2.81 |
| 2 | LLaVA-Video-7B-Qwen2 | 2.77 |
| 3 | VideoLLaMA3-7B | 2.55 |
| 4 | VideoLLaMA3-2B | 2.49 |
| 5 | LLavA-OV-Qwen2-0.5B | 2.43 |
| 6 | LLaVA-Next-7B | 2.28 |
| 7 | Qwen2.5-VL-3B | 2.26 |
| 8 | InternVL2.5-2B | 1.68 |

Table 9: Temporal QA

| Rank | Model | Acc |
|------|-------|-----|
| 1 | LLaVA-Video-7B-Qwen2 | 2.71 |
| 2 | LLavA-OV-Qwen2-7B | 2.66 |
| 3 | LLaVA-Next-7B | 2.48 |
| 4 | VideoLLaMA3-7B | 2.45 |
| 5 | LLavA-OV-Qwen2-0.5B | 2.45 |
| 6 | VideoLLaMA3-2B | 2.44 |
| 7 | Qwen2.5-VL-3B | 2.31 |
| 8 | InternVL2.5-2B | 1.95 |

Table 10: Short Temporal

| Rank | Model | Acc |
|------|-------|-----|
| 1 | LLaVA-Video-7B-Qwen2 | 3.51 |
| 2 | LLavA-OV-Qwen2-7B | 3.34 |
| 3 | LLavA-OV-Qwen2-0.5B | 3.12 |
| 4 | VideoLLaMA3-7B | 3.06 |
| 5 | VideoLLaMA3-2B | 2.99 |
| 6 | LLaVA-Next-7B | 2.95 |
| 7 | Qwen2.5-VL-3B | 2.75 |
| 8 | InternVL2.5-2B | 1.89 |

Table 11: Spatial QA

| Rank | Model | Acc |
|------|-------|-----|
| 1 | LLaVA-Video-7B-Qwen2 | 3.05 |
| 2 | LLavA-OV-Qwen2-7B | 2.89 |
| 3 | LLaVA-Next-7B | 2.81 |
| 4 | VideoLLaMA3-7B | 2.78 |
| 5 | Qwen2.5-VL-3B | 2.70 |
| 6 | LLavA-OV-Qwen2-0.5B | 2.69 |
| 7 | VideoLLaMA3-2B | 2.66 |
| 8 | InternVL2.5-2B | 2.26 |

Table 12: Reasoning QA

| Rank | Model | Avg |
|------|-------|-----|
| 1 | LLavA-OV-Qwen2-0.5B | 2.94 |
| 2 | LLavA-OV-Qwen2-7B | 2.53 |
| 3 | LLaVA-Next-7B | 2.32 |
| 4 | VideoLLaMA3-7B | 2.03 |
| 5 | LLavA-Video-7B-Qwen2 | 1.92 |
| 6 | InternVL2.5-2B | 1.88 |
| 7 | VideoLLaMA3-2B | 1.87 |
| 8 | Qwen2.5-VL-3B | 1.85 |

Table 13: Anomaly Detection

| Rank | Model | Avg |
|------|-------|-----|
| 1 | LLavA-OV-Qwen2-7B | 2.56 |
| 2 | LLaVA-Video-7B-Qwen2 | 2.59 |
| 3 | LLaVA-Next-7B | 2.56 |
| 4 | LLavA-OV-Qwen2-0.5B | 2.47 |
| 5 | Qwen2.5-VL-3B | 2.19 |
| 6 | VideoLLaMA3-2B | 2.03 |
| 7 | VideoLLaMA3-7B | 1.53 |
| 8 | InternVL2.5-2B | 1.12 |

Table 14: Anomaly Classification

| Rank | Model | Avg |
|------|-------|-----|
| 1 | LLaVA-Video-7B-Qwen2 | 2.76 |
| 2 | LLavA-OV-Qwen2-0.5B | 2.69 |
| 3 | LLaVA-Next-7B | 2.68 |
| 4 | VideoLLaMA3-7B | 2.66 |
| 5 | LLaVA-Next-7B | 2.59 |
| 6 | VideoLLaMA3-2B | 2.46 |
| 7 | Qwen2.5-VL-3B | 2.24 |
| 8 | InternVL2.5-2B | 1.46 |

Table 15: Anomaly Subject

| Rank | Model | Acc |
|------|-------|-----|
| 1 | LLaVA-Video-7B-Qwen2 | 2.76 |
| 2 | LLavA-OV-Qwen2-7B | 2.54 |
| 3 | LLavA-OV-Qwen2-0.5B | 2.43 |
| 4 | LLaVA-Next-7B | 2.11 |
| 5 | VideoLLaMA3-7B | 2.01 |
| 6 | VideoLLaMA3-2B | 1.89 |
| 7 | Qwen2.5-VL-3B | 1.74 |
| 8 | $InternVL2_5 - 2B$ | 0.55 |

Table 16: Anomaly Description

| Rank | Model | Acc |
|------|-------|-----|
| 1 | LLaVA-Next-7B | 1.96 |
| 2 | VideoLLaMA3-7B | 1.75 |
| 3 | LLavA-OV-Qwen2-0.5B | 1.67 |
| 4 | LLavA-Video-7B-Qwen2 | 1.52 |
| 5 | LLavA-OV-Qwen2-7B | 1.46 |
| 6 | VideoLLaMA3-2B | 1.37 |
| 7 | Qwen2.5-VL-3B | 1.32 |
| 8 | InternVL2.5-2B | 0.64 |

Table 17: Anomaly Cause

| Rank | Model | Acc |
|------|-------|-----|
| 1 | LLaVA-Video-7B-Qwen2 | 1.81 |
| 2 | LLaVA-Next-7B | 1.65 |
| 3 | LLavA-OV-Qwen2-0.5B | 1.60 |
| 4 | LLavA-OV-Qwen2-7B | 1.55 |
| 5 | VideoLLaMA3-7B | 1.27 |
| 6 | VideoLLaMA3-2B | 1.18 |
| 7 | Qwen2.5-VL-3B | 1.13 |
| 8 | InternVL2.5-2B | 0.74 |

Table 18: Anomaly Result

Figure 11: Performance rankings of LVLMs across different QA tasks. No.1-6 lists denote normal video clips vs. normal QA tasks. No.7-12 lists denote abnormal video clips vs. normal QA tasks. No.13-18 lists denote abnormal video clips vs. abnormal QA tasks.

## E.2 More results with Five Metrics with Normal QA Tasks

More results with five metrics including CI, DO, CU, TU, AVG, across normal QA tasks on normal video clips have been shown in Table 8. From the experimental results, it can be seen that large models have significant advantages in most tasks, especially the fine-tuned versions can further enhance performance. Specifically, in tasks such as SummaryQA, Generic QA and Temporal QA, LLaVA-OV-Qwen2-7B-ov and LLaVA-Video-7B achieved optimal or near-optimal results, demonstrating strong cross-modal understanding and reasoning capabilities. In the Spatial QA and Reasoning QA tasks, large models (such as LLaVA-Video-7B and LLaVA-Video-7B[†]) also performed outstandingly, indicating that they have stronger generalization ability in complex spatial relationships and reasoning scenarios. In contrast, small-scale models (such as LLaVA-OV-Qwen2-0.5B and InternVL-2.5-2B) performed poorly in all tasks, especially lagging significantly in SummaryQA and Temporal QA. Overall, the results verify the significant improvement effect of model scale and targeted fine-tuning on the performance of video question answering tasks, and also highlight the differentiated requirements for model capabilities in different tasks during evaluation.

Table 8: Performance of different vision-language models across normal QA tasks on normal video clips. All values are shown in blue, with the highest value in each row bolded. [†] represents our finetuned LVLMs.

| Task | Metric | LLaVA-OV-Qwen2-0.5B | InternVL 2.5-2B | VideoLLaMA3-2B | Qwen2.5-VL-3B-Instruct | LLaVA-Next-7B | LLaVA-OV-Qwen2-7B-ov | LLaVA-Video-7B | VideoLLaMA3-7B | Qwen2.5-VL-3B-Instruct[†] | LLaVA-Video-7B[†] |
|---|---|---|---|---|---|---|---|---|---|---|---|
| | CI | 2.95 | 0.45 | 2.65 | 2.29 | 2.32 | **3.34** | 3.30 | 2.82 | 2.77 | 3.20 |
| | DO | 2.63 | 0.48 | 2.35 | 2.12 | 2.08 | **2.92** | 2.78 | 2.62 | 2.53 | 2.90 |
| SummaryQA | CU | 2.88 | 0.75 | 2.64 | 2.36 | 2.34 | 3.21 | **3.22** | 2.79 | 2.73 | 3.09 |
| | TU | 2.61 | 0.55 | 2.33 | 2.03 | 1.93 | **2.91** | **2.91** | 2.51 | 2.45 | 2.84 |
| | Avg. | 2.76 | 0.56 | 2.49 | 2.20 | 2.17 | **3.10** | 3.05 | 2.68 | 2.62 | 3.01 |
| | CI | 2.97 | 2.27 | 3.02 | 2.83 | 3.11 | 3.27 | 3.19 | 3.20 | 2.92 | **3.34** |
| | DO | 2.60 | 2.13 | 2.70 | 2.56 | 2.86 | 2.91 | 2.79 | 2.87 | 2.67 | **3.02** |
| Generic QA | CU | 2.90 | 2.32 | 2.94 | 2.77 | 3.03 | 3.17 | 3.10 | 3.11 | 2.85 | **3.25** |
| | TU | 2.65 | 2.06 | 2.70 | 2.50 | 2.73 | 2.94 | 2.88 | 2.87 | 2.58 | **3.03** |
| | Avg. | 2.78 | 2.20 | 2.84 | 2.66 | 2.93 | 3.08 | 2.99 | 3.01 | 2.76 | **3.16** |
| | CI | 2.73 | 1.83 | 2.84 | 2.77 | 2.68 | 3.08 | 2.98 | 2.91 | 2.97 | **3.18** |
| | DO | 2.52 | 1.90 | 2.66 | 2.61 | 2.50 | 2.85 | 2.75 | 2.70 | 2.80 | **2.95** |
| Temporal QA | CU | 2.73 | 2.03 | 2.83 | 2.75 | 2.67 | 3.03 | 2.95 | 2.89 | 2.92 | **3.12** |
| | TU | 2.48 | 1.80 | 2.61 | 2.51 | 2.38 | 2.84 | 2.74 | 2.72 | 2.68 | **2.95** |
| | Avg. | 2.62 | 1.89 | 2.73 | 2.66 | 2.56 | 2.95 | 2.85 | 2.81 | 2.84 | **3.05** |
| | CI | 2.71 | 2.04 | 2.73 | 2.79 | 2.70 | 2.92 | 2.90 | 2.76 | 3.04 | **3.17** |
| | DO | 2.48 | 2.00 | 2.45 | 2.66 | 2.67 | 2.63 | 2.65 | 2.46 | 2.93 | **2.94** |
| Short Temporal | CU | 2.76 | 2.27 | 2.74 | 2.80 | 2.78 | 2.92 | 2.94 | 2.79 | 3.00 | **3.18** |
| | TU | 2.46 | 2.00 | 2.51 | 2.52 | 2.50 | 2.68 | 2.65 | 2.56 | 2.75 | **2.98** |
| | Avg. | 2.60 | 2.08 | 2.61 | 2.69 | 2.66 | 2.79 | 2.78 | 2.65 | 2.93 | **3.07** |
| | CI | 3.22 | 1.90 | 3.07 | 2.94 | 3.03 | 3.41 | **3.60** | 3.16 | 2.97 | 3.52 |
| | DO | 2.96 | 1.83 | 2.80 | 2.78 | 2.88 | 3.17 | **3.30** | 2.91 | 2.85 | 3.28 |
| Spatial QA | CU | 3.23 | 2.10 | 3.10 | 2.97 | 3.06 | 3.41 | **3.57** | 3.18 | 2.98 | 3.51 |
| | TU | 3.02 | 1.85 | 2.89 | 2.74 | 2.83 | 3.23 | **3.40** | 2.98 | 2.76 | 3.34 |
| | Avg. | 3.10 | 1.92 | 2.97 | 2.86 | 2.95 | 3.31 | **3.47** | 3.06 | 2.89 | 3.41 |
| | CI | 3.05 | 2.45 | 3.02 | 3.11 | 3.19 | 3.22 | 3.39 | 3.13 | 3.14 | **3.45** |
| | DO | 2.75 | 2.29 | 2.69 | 2.94 | 2.97 | 2.87 | 3.05 | 2.80 | 3.06 | **3.15** |
| Reasoning QA | CU | 3.07 | 2.57 | 3.05 | 3.13 | 3.19 | 3.23 | 3.38 | 3.15 | 3.14 | **3.44** |
| | TU | 2.84 | 2.32 | 2.81 | 2.85 | 2.90 | 3.00 | 3.18 | 2.91 | 2.90 | **3.25** |
| | Avg. | 2.93 | 2.41 | 2.89 | 3.01 | 3.06 | 3.08 | 3.25 | 2.99 | 3.06 | **3.32** |

More results with five metrics including CI, DO, CU, TU, AVG, across normal QA tasks on abnormal video clips have been shown in Table 9. Overall, large-scale and specially fine-tuned models maintain the lead in most tasks, but abnormal scenarios significantly lower the overall performance, especially for Summary QA problems. In Spatial QA, LLaVA-Video-7B and its fine-tuned version achieved the highest average scores, respectively, in the entire table, demonstrating a robust ability to understand spatial relationships. In the Reasoning QA and Short Temporal tasks, the fine-tuned LLAVA-Video-7b-fine-tuning ranked first, respectively, indicating that fine-tuning has significant gains in complex reasoning and short-sequence understanding. In Temporal QA and Generic QA, this model also achieved the highest or tied highest scores, forming the first echelon with the unfine-tuned large models. In contrast, Summary QA was most affected by anomalies. Small-scale models are at the bottom of all sub-tasks and metrics, while Qwen2.5-VL-3B-Instruct[†] shows stable improvement in multiple tasks compared to the unfine-tuned version. But it is still difficult to catch up with 7B-level models. In summary, the results under abnormal video conditions highlight the importance of model

Table 9: Performance of different vision-language models across normal QA tasks on abnormal video clips. All values are shown in green, with the highest value in each row bolded. † represents our finetuned LVLMs.

| Task | Metric | LLaVA-OV-Qwen2-0.5B | InternVL 2.5-2B | VideoLLaMA3-2B | Qwen2.5-VL-3B-Instruct | LLaVA-Next-7B | LLaVA-OV-Qwen2-7B-ov | LLaVA-Video-7B | VideoLLaMA3-7B | Qwen2.5-VL-3B-Instruct† | LLaVA-Video-7B† |
|------|--------|------|------|------|------|------|------|------|------|------|------|
| Summary QA | CI | 2.56 | 0.22 | 2.06 | 1.46 | 1.78 | 2.97 | **3.03** | 2.35 | 2.05 | 2.76 |
| | DO | 2.35 | 0.26 | 1.92 | 1.46 | 1.68 | **2.69** | 2.63 | 2.28 | 1.96 | 2.58 |
| | CU | 2.56 | 0.60 | 2.17 | 1.68 | 1.90 | 2.90 | **3.02** | 2.43 | 2.14 | 2.73 |
| | TU | 2.24 | 0.40 | 1.84 | 1.38 | 1.52 | 2.61 | **2.72** | 2.11 | 1.82 | 2.45 |
| | Avg. | 2.43 | 0.37 | 2.00 | 1.49 | 1.72 | 2.79 | **2.85** | 2.29 | 1.99 | 2.63 |
| Generic QA | CI | 2.66 | 1.95 | 2.68 | 2.23 | 2.72 | 3.01 | 2.96 | 2.83 | 2.43 | **3.07** |
| | DO | 2.41 | 1.90 | 2.48 | 2.15 | 2.58 | 2.74 | 2.63 | 2.63 | 2.35 | **2.86** |
| | CU | 2.65 | 2.08 | 2.67 | 2.29 | 2.71 | 2.97 | 2.94 | 2.81 | 2.47 | **3.03** |
| | TU | 2.37 | 1.80 | 2.39 | 2.01 | 2.39 | 2.71 | 2.68 | 2.53 | 2.18 | **2.79** |
| | Avg. | 2.52 | 1.93 | 2.55 | 2.17 | 2.60 | 2.86 | 2.80 | 2.70 | 2.36 | **2.94** |
| Temporal QA | CI | 2.50 | 1.57 | 2.53 | 2.26 | 2.31 | 2.89 | 2.86 | 2.61 | 2.51 | **2.93** |
| | DO | 2.38 | 1.69 | 2.47 | 2.27 | 2.29 | 2.74 | 2.68 | 2.50 | 2.52 | **2.78** |
| | CU | 2.57 | 1.84 | 2.59 | 2.37 | 2.40 | 2.90 | 2.88 | 2.65 | 2.57 | **2.93** |
| | TU | 2.29 | 1.60 | 2.34 | 2.12 | 2.10 | 2.69 | 2.65 | 2.44 | 2.33 | **2.73** |
| | Avg. | 2.43 | 1.68 | 2.49 | 2.26 | 2.28 | 2.81 | 2.77 | 2.55 | 2.48 | **2.85** |
| Short Temporal | CI | 2.51 | 1.89 | 2.52 | 2.33 | 2.48 | 2.75 | 2.77 | 2.53 | 2.61 | **2.92** |
| | DO | 2.36 | 1.89 | 2.30 | 2.32 | 2.51 | 2.52 | 2.60 | 2.30 | 2.65 | **2.78** |
| | CU | 2.61 | 2.15 | 2.60 | 2.44 | 2.63 | 2.82 | 2.87 | 2.61 | 2.70 | **2.99** |
| | TU | 2.31 | 1.88 | 2.35 | 2.15 | 2.30 | 2.53 | 2.60 | 2.37 | 2.40 | **2.77** |
| | Avg. | 2.45 | 1.95 | 2.44 | 2.31 | 2.48 | 2.66 | 2.71 | 2.45 | 2.59 | **2.86** |
| Spatial QA | CI | 3.23 | 1.85 | 3.09 | 2.83 | 3.03 | 3.45 | **3.65** | 3.17 | 3.00 | 3.60 |
| | DO | 2.98 | 1.79 | 2.83 | 2.68 | 2.89 | 3.20 | **3.33** | 2.91 | 2.87 | 3.33 |
| | CU | 3.23 | 2.09 | 3.11 | 2.86 | 3.06 | 3.43 | **3.62** | 3.19 | 2.97 | 3.58 |
| | TU | 3.04 | 1.82 | 2.92 | 2.64 | 2.83 | 3.27 | **3.45** | 2.99 | 2.76 | 3.40 |
| | Avg. | 3.12 | 1.89 | 2.99 | 2.75 | 2.95 | 3.34 | **3.51** | 3.06 | 2.89 | 3.48 |
| Reasoning QA | CI | 2.79 | 2.26 | 2.74 | 2.75 | 2.92 | 3.00 | 3.15 | 2.86 | 2.87 | **3.23** |
| | DO | 2.53 | 2.16 | 2.48 | 2.65 | 2.72 | 2.70 | 2.87 | 2.60 | 2.86 | **2.99** |
| | CU | 2.83 | 2.43 | 2.84 | 2.84 | 2.95 | 3.04 | 3.21 | 2.95 | 3.00 | **3.29** |
| | TU | 2.60 | 2.17 | 2.57 | 2.57 | 2.64 | 2.81 | 2.98 | 2.70 | 2.72 | **3.08** |
| | Avg. | 2.69 | 2.26 | 2.66 | 2.70 | 2.81 | 2.89 | 3.05 | 2.78 | 2.85 | **3.15** |

scale as a strong performance baseline. This indicates that fine-tuning for the target domain can bring significant benefits in temporal and inference tasks, but the improvement in cross-shot integration and generalization capabilities is still limited. This provides a direction for strengthening long-term dependency modeling and robust semantic extraction in abnormal scenarios in the future.

## E.3 MORE RESULTS WITH FIVE METRICS WITH ABNORMAL QA TASKS

More results with five metrics including CI, DO, CU, TU, AVG, across abnormal QA tasks on abnormal video clips have been shown in Table 10. Overall, the 7b-level model and its fine-tuned version have an advantage. Among them, LLAVA-Video-7b-fine-tune achieves a significant lead in Detection QA. And it also ranked in the first echelon in both Classification QA and Subject QA; In contrast, the unfine-tuned LLaVA-Video-7B performed best in Description QA and Result QA, while LLaVA-Next-7B gave the highest average score in Cause QA. It indicates that the ability of "causal explanation" does not always benefit the most from the specific fine-tuning of videos. The trends among the various metrics are basically consistent: Recognition and enumeration subtasks (Detection/Classification/Subject) benefit fine-tuning more significantly, while cross-fragment integration and causal inference (Description/Cause/Result) impose higher requirements on the structured representation and temporal-logical connection of the model. The advantages of fine-tuning are relatively convergent. Small-scale models (such as 0.5B / 2B scale) are generally weak in all tasks and metrics, further confirming the crucial role of model scale and target domain fine-tuning in understanding abnormal scenarios. It also suggests that future work can enhance long-term dependency modeling and more refined event structure learning in causal and outcome reasoning to narrow the performance gap with recognition tasks.

## E.4 COMPARISONS OF MODEL PERFORMANCE ON VARIOUS VIDEO EVENT CATEGORIES

In our dataset, certain abnormal event categories such as Threat and Water Incident contain relatively few samples (in single digits shown in Table 5). Rather than excluding these categories, we inten-

Table 10: Performance of different vision-language models across abnormal QA tasks on abnormal video clips. All values are shown in brown, with the highest value in each row bolded. [†] represents our finetuned LVLMs.

| Task | Metric | LLaVA-OV-Qwen2-0.5B | InternVL2.5-2B | VideoLLaMA3-2B | Qwen2.5-VL-3B-Instruct | LLaVA-Next-7B | LLaVA-OV-Qwen2-7B-ov | LLaVA-Video-7B | VideoLLaMA3-7B | Qwen2.5-VL-3B-Instruct[†] | LLaVA-Video-7B[†] |
|---|---|---|---|---|---|---|---|---|---|---|---|
| Detection QA | CI | 3.10 | 1.97 | 1.90 | 1.92 | 2.39 | 2.54 | 1.95 | 2.07 | 1.57 | **4.57** |
| | DO | 2.69 | 1.71 | 1.81 | 1.93 | 2.51 | 2.50 | 1.85 | 1.97 | 1.75 | **4.29** |
| | CU | 3.11 | 2.00 | 1.94 | 1.88 | 2.29 | 2.54 | 1.96 | 2.11 | 1.54 | **4.62** |
| | TU | 2.86 | 1.83 | 1.81 | 1.69 | 2.09 | 2.52 | 1.90 | 1.95 | 1.45 | **4.34** |
| | Avg. | 2.94 | 1.88 | 1.87 | 1.85 | 2.32 | 2.53 | 1.92 | 2.03 | 1.58 | **4.46** |
| Classification QA | CI | 2.36 | 0.86 | 1.88 | 2.00 | 2.47 | 2.52 | 2.47 | 1.37 | 1.90 | **3.26** |
| | DO | 2.53 | 1.22 | 2.07 | 2.40 | 2.74 | 2.61 | 2.72 | 1.49 | 2.42 | **3.25** |
| | CU | 2.55 | 1.36 | 2.22 | 2.30 | 2.63 | 2.77 | 2.71 | 1.84 | 2.18 | **3.41** |
| | TU | 2.43 | 1.05 | 1.94 | 2.05 | 2.41 | 2.48 | 2.45 | 1.41 | 2.01 | **3.13** |
| | Avg. | 2.47 | 1.12 | 2.03 | 2.19 | 2.56 | 2.60 | 2.59 | 1.53 | 2.13 | **3.26** |
| Subject QA | CI | 2.76 | 1.30 | 2.49 | 2.24 | 2.69 | 2.75 | 2.83 | 2.71 | 2.55 | **3.31** |
| | DO | 2.52 | 1.46 | 2.32 | 2.23 | 2.56 | 2.51 | 2.59 | 2.53 | 2.69 | **3.21** |
| | CU | 2.87 | 1.74 | 2.64 | 2.43 | 2.75 | 2.86 | 2.93 | 2.83 | 2.66 | **3.40** |
| | TU | 2.62 | 1.36 | 2.38 | 2.07 | 2.36 | 2.62 | 2.71 | 2.55 | 2.35 | **3.17** |
| | Avg. | 2.69 | 1.46 | 2.46 | 2.24 | 2.59 | 2.68 | 2.76 | 2.66 | 2.56 | **3.27** |
| Description QA | CI | 2.56 | 0.35 | 1.98 | 1.76 | 2.21 | 2.71 | **2.92** | 2.05 | 2.11 | 2.79 |
| | DO | 2.28 | 0.46 | 1.73 | 1.68 | 2.06 | 2.40 | 2.58 | 1.90 | 2.03 | **2.66** |
| | CU | 2.61 | 0.81 | 2.10 | 1.88 | 2.27 | 2.67 | **2.92** | 2.23 | 2.17 | 2.81 |
| | TU | 2.27 | 0.55 | 1.76 | 1.64 | 1.92 | 2.39 | **2.62** | 1.88 | 1.91 | 2.53 |
| | Avg. | 2.43 | 0.55 | 1.89 | 1.74 | 2.11 | 2.54 | **2.76** | 2.01 | 2.06 | 2.70 |
| Cause QA | CI | 1.69 | 0.46 | 1.31 | 1.29 | **2.03** | 1.51 | 1.56 | 1.69 | 1.64 | 1.81 |
| | DO | 1.49 | 0.55 | 1.23 | 1.28 | **1.87** | 1.16 | 1.24 | 1.71 | 1.69 | 1.62 |
| | CU | 1.87 | 0.87 | 1.55 | 1.40 | **2.06** | 1.66 | 1.72 | 1.86 | 1.79 | 1.98 |
| | TU | 1.62 | 0.69 | 1.38 | 1.29 | **1.86** | 1.51 | 1.54 | 1.71 | 1.63 | 1.75 |
| | Avg. | 1.67 | 0.64 | 1.37 | 1.32 | **1.96** | 1.46 | 1.52 | 1.75 | 1.69 | 1.79 |
| Result QA | CI | 1.62 | 0.53 | 1.20 | 1.08 | 1.73 | 1.63 | 1.89 | 1.25 | 1.25 | **1.96** |
| | DO | 1.39 | 0.63 | 0.91 | 1.03 | 1.49 | 1.22 | 1.56 | 0.99 | 1.19 | **1.77** |
| | CU | 1.84 | 0.99 | 1.47 | 1.35 | 1.87 | 1.85 | 2.07 | 1.56 | 1.53 | **2.15** |
| | TU | 1.54 | 0.80 | 1.15 | 1.09 | 1.51 | 1.51 | 1.74 | 1.26 | 1.24 | **1.85** |
| | Avg. | 1.60 | 0.74 | 1.18 | 1.13 | 1.65 | 1.55 | 1.81 | 1.27 | 1.30 | **1.93** |

tionally retained them to preserve the diversity of event types and to better reflect the generalization capabilities of LVLMs across heterogeneous semantic scenarios. This design choice ensures that the benchmark captures not only frequent but also rare, yet semantically important, abnormal events. It is worth noting that the overall evaluation metrics reported in the main paper are aggregated without assigning additional weights to these low-frequency categories. As a result, their contribution to the final performance results remains limited.

Furthermore, the comparison results across different categories demonstrate that the performance of models on rare categories does not systematically deviate from their performance on more common ones. This indicates that the inclusion of such categories enhances coverage without introducing undue bias. The detailed results are shown in Table 11.

## E.5 Results of Locally-deployed and API-called LVLMs on Anomalous Videos

Table 12 compares finetuned locally deployed LVLM models (e.g., Qwen2.5-VL-3B-Instruct, LLaVA-Video-7B, and fine-tuned models) against API-called models (Baidu ERNIE 4.5 Turbo VL Baidu (2025) , Gemini 2.5 Pro Google (2025) , GPT-4o OpenAI (2024) , InternVL3.5 Wang et al. (2025) ) on anomalous video tasks. Across most of the subtasks, API models show higher average scores than local models, with GPT-4o OpenAI (2024) and InternVL3.5 Wang et al. (2025) often leading, highlighting API models' superior performance in accuracy and detail. However, finetuned LLaVa-series models also show competitive performance.

Table 11: Model Performance on Various Environment Categories

| Category | LLaVA-OV-Qwen2-0.5B | InternVL 2.5-2B | VideoLLaMA 3-2B | Qwen2.5-VL-3B-Instruct | LLaVA-NeXT-7B | LLaVA-OV-Qwen2-7B | LLaVA-Video-7B | VideoLLaMA 3-7B | Avg. |
|---|---|---|---|---|---|---|---|---|---|
| normal | 2.8 | 1.84 | 2.76 | 2.69 | 2.73 | 3.05 | **3.07** | 2.87 | 2.73 |
| Abuse | 2.25 | 1.42 | 1.92 | 1.68 | 2.19 | 2.26 | **2.37** | 2.01 | 2.01 |
| Arrest | 2.61 | 1.62 | 2.15 | 1.98 | 2.46 | 2.60 | **2.62** | 2.24 | 2.28 |
| Assault | 2.41 | 1.40 | 2.11 | 1.89 | 2.24 | 2.55 | **2.59** | 2.20 | 2.17 |
| Burglary | 2.44 | 1.62 | 2.22 | 2.00 | 2.34 | 2.60 | **2.67** | 2.21 | 2.27 |
| Pursuit | **2.85** | 1.70 | 1.97 | 2.09 | 2.09 | 2.81 | 2.40 | 2.05 | 2.24 |
| Explosion | 2.74 | 1.13 | 2.26 | 1.99 | 2.47 | 2.87 | **2.90** | 2.38 | 2.34 |
| Fighting | 2.62 | 1.54 | 2.19 | 2.11 | 2.59 | **2.73** | 2.67 | 2.35 | 2.35 |
| Fire | 2.65 | 1.33 | 2.33 | 2.21 | 2.63 | 2.63 | **2.67** | 2.47 | 2.36 |
| Object Falling | 2.42 | 1.47 | 2.08 | 1.90 | 2.22 | 2.53 | **2.69** | 2.24 | 2.19 |
| People Falling | 2.48 | 1.45 | 2.20 | 2.00 | 2.36 | 2.55 | **2.58** | 2.29 | 2.24 |
| Robbery | 2.34 | 1.27 | 2.11 | 2.07 | 2.25 | 2.55 | **2.60** | 2.25 | 2.18 |
| Shooting | 2.28 | 1.51 | 2.32 | 2.01 | 2.08 | 2.40 | **2.54** | 2.30 | 2.18 |
| Shoplifting | 2.20 | 1.29 | 1.96 | 1.85 | 2.20 | 2.27 | **2.39** | 2.00 | 2.02 |
| Stealing | 2.36 | 1.26 | 2.03 | 1.91 | 2.19 | 2.45 | **2.46** | 2.17 | 2.10 |
| Threatening | **2.55** | 1.68 | 1.97 | 2.06 | 2.48 | 2.52 | 2.15 | 2.16 | 2.20 |
| Traffic Accident | 2.47 | 1.21 | 2.26 | 2.16 | 2.39 | 2.56 | **2.61** | 2.32 | 2.25 |
| Vandalism | 2.33 | 1.34 | 2.05 | 1.93 | 2.26 | 2.44 | **2.49** | 2.12 | 2.12 |
| Water Incident | 3.13 | 1.44 | 2.89 | **3.36** | 3.23 | 2.81 | 3.06 | 3.29 | 2.90 |

Table 12: Local-deployed LVLMs and API-called LVLMs(Baidu ERNIE Baidu (2025) , Gemini 2.5 Pro Google (2025) , GPT-4o OpenAI (2024) , InternVL3.5 Wang et al. (2025) ) on Anomalous Videos. $^{\dagger}$ represents our finetuned LVLMs.

| Task | Metric | Qwen2.5-VL-3B-Instruct | Qwen2.5-VL-3B-Instruct$^{\dagger}$ | LLaVA-Video-7B-Qwen2 | LLaVA-Video-7B-Qwen2$^{\dagger}$ | Baidu ERNIE 4.5 Turbo VL | Gemini 2.5 Pro | GPT-4o | InternVL3.5 |
|---|---|---|---|---|---|---|---|---|---|
| Detection QA | CI | 1.92 | 1.57 | 1.95 | **4.57** | 3.28 | 4.55 | 3.77 | 3.74 |
| | DO | 1.93 | 1.75 | 1.85 | **4.29** | 3.20 | 4.42 | 3.59 | 3.67 |
| | CU | 1.88 | 1.54 | 1.96 | **4.62** | 3.26 | 4.50 | 3.62 | 3.58 |
| | TU | 1.69 | 1.45 | 1.90 | **4.34** | 3.19 | 4.39 | 3.35 | 3.17 |
| | Avg. | 1.85 | 1.58 | 1.92 | **4.46** | 3.23 | 4.47 | 3.58 | 3.54 |
| Classification QA | CI | 2.00 | 1.90 | 2.47 | 3.26 | 2.75 | 3.51 | 3.45 | **3.80** |
| | DO | 2.40 | 2.42 | 2.72 | 3.25 | 2.72 | 3.55 | 3.70 | **3.95** |
| | CU | 2.30 | 2.18 | 2.71 | 3.41 | 3.00 | 3.57 | 3.63 | **3.78** |
| | TU | 2.05 | 2.01 | 2.45 | 3.13 | 2.69 | 3.43 | 3.34 | **3.55** |
| | Avg. | 2.19 | 2.13 | 2.59 | 3.26 | **2.79** | 3.51 | 3.53 | 3.77 |
| Subject QA | CI | 2.24 | 2.55 | 2.83 | 3.31 | 2.49 | 2.98 | 2.54 | **3.35** |
| | DO | 2.23 | 2.69 | 2.59 | 3.21 | 2.39 | 2.99 | 2.48 | **3.34** |
| | CU | 2.43 | 2.66 | 2.93 | **3.40** | 2.63 | 3.00 | 2.71 | 3.32 |
| | TU | 2.07 | 2.35 | 2.71 | **3.17** | 2.39 | 2.79 | 2.35 | 3.03 |
| | Avg. | 2.24 | 2.56 | 2.76 | **3.27** | 2.47 | 2.94 | 2.52 | 3.26 |
| Description QA | CI | 1.76 | 2.11 | 2.92 | 2.79 | 2.28 | 2.85 | 3.02 | **3.12** |
| | DO | 1.68 | 2.03 | 2.58 | 2.66 | 2.14 | 2.64 | 2.78 | **2.90** |
| | CU | 1.88 | 2.17 | 2.92 | 2.81 | 2.35 | 2.86 | 3.00 | **3.09** |
| | TU | 1.64 | 1.91 | 2.62 | 2.53 | 2.07 | 2.57 | 2.72 | **2.81** |
| | Avg. | 1.74 | 2.06 | 2.76 | 2.70 | 2.21 | 2.73 | 2.88 | **2.98** |
| Cause QA | CI | 1.29 | 1.64 | 1.56 | 1.81 | 1.63 | 2.05 | 2.24 | **3.03** |
| | DO | 1.28 | 1.69 | 1.24 | 1.62 | 1.51 | 2.00 | 2.13 | **2.81** |
| | CU | 1.40 | 1.79 | 1.72 | 1.98 | 1.80 | 2.05 | 2.33 | **2.93** |
| | TU | 1.29 | 1.63 | 1.54 | 1.75 | 1.61 | 1.87 | 2.09 | **2.71** |
| | Avg. | 1.32 | 1.69 | 1.52 | 1.79 | 1.64 | 1.99 | 2.20 | **2.87** |
| Result QA | CI | 1.08 | 1.25 | 1.89 | 1.96 | 1.25 | 2.23 | 2.52 | **3.09** |
| | DO | 1.03 | 1.19 | 1.56 | 1.77 | 1.01 | 2.04 | 2.26 | **2.74** |
| | CU | 1.35 | 1.53 | 2.07 | 2.15 | 1.54 | 2.28 | 2.53 | **2.98** |
| | TU | 1.09 | 1.24 | 1.74 | 1.85 | 1.25 | 2.04 | 2.22 | **2.62** |
| | Avg. | 1.13 | 1.30 | 1.81 | 1.93 | 1.26 | 2.15 | 2.38 | **2.86** |

## E.6 ANALYZE THE ENVIRONMENT OF THE SURVEILLANCE VIDEO

We have made additional statistical information analysis regarding the surveillance environments in our abnormal video dataset. Specifically, approximately 51% of the clips are from indoor scenes and 49% from outdoor scenes. Day and night scenarios are similarly balanced, each accounting for

roughly half of the data. As for occlusion, around 83% of the clips contain no significant occlusion, while the remaining 17% involve partial or full occlusion, resulting in a ratio of approximately 5:1.

To further address this point, we provide evaluation results of various LVLMs under different surveillance conditions in the following Table 13. The performance differences observed across these environmental settings are relatively small, indicating that the evaluated models demonstrate a stable level of cognitive understanding regardless of scene variation. This robustness can be attributed to two main factors: (1) the LVLMs have been pre-trained on diverse and complex visual inputs, enabling them to generalize effectively across a wide range of surveillance scenarios; and (2) the video clips in our dataset are sourced from publicly available, high-quality datasets such as MSAD, MEVA, NWPU, and UCF-Crime. These datasets generally feature clear and well-lit footage, including night, and contain relatively few examples of extreme occlusion.

Table 13: Results of different video experiment conditions.

| Condition | Proportion | LLaVA-OV-0.5B | InternVL2.5-2B | VideoLLaMA3-2B | Qwen2.5-VL-3B | LLaVA-NeXT-7B | LLaVA-OV-7B | LLaVA-Video-7B | VideoLLaMA3-7B | Avg. | Std. |
|-----------|-----------|---------------|----------------|----------------|---------------|---------------|-------------|----------------|----------------|------|------|
| Indoor | 51% | 2.48 | 1.56 | 2.29 | 2.07 | 2.37 | 2.70 | **2.77** | 2.40 | 2.33 | 0.36 |
| Outdoor | 49% | 2.58 | 1.49 | 2.39 | 2.23 | 2.44 | 2.75 | **2.77** | 2.49 | 2.40 | 0.38 |
| Unoccluded | 83% | 2.53 | 1.51 | 2.35 | 2.17 | 2.40 | 2.72 | **2.76** | 2.45 | 2.37 | 0.37 |
| Occluded | 17% | 2.54 | 1.57 | 2.32 | 2.05 | 2.44 | 2.76 | **2.81** | 2.43 | 2.36 | 0.37 |
| Day | 57% | 2.51 | 1.51 | 2.35 | 2.19 | 2.41 | 2.70 | **2.75** | 2.46 | 2.36 | 0.36 |
| Night | 43% | 2.56 | 1.54 | 2.33 | 2.09 | 2.40 | 2.75 | **2.79** | 2.43 | 2.36 | 0.38 |

## F    VISUALIZATION FOR INFERENCE RESULTS OF LVLMS.

Figure 12, Figure 13, Figure 14, and Figure 15 show the failed cases in the causal reasoning QA inference. We have given the model's answers and error analysis in these figures. The analysis of the cause and result QA pairs across the LVLMs reveals several recurring issues in the performance of vision-language models (LVLMs) when interpreting abnormal video events. These issues can be grouped into the following categories, as shown in Table 14.

These examples highlight common issues related to event identification, causal reasoning, generalization, and language generation. These models struggle to accurately interpret complex video events, often providing vague, incomplete, or incorrect responses. While some models are better at identifying abnormal events or constructing causal chains, the overall performance remains limited by these recurring errors. To improve the robustness and reliability of LVLMs, future research should focus on enhancing their ability to correctly identify and reason about abnormal video events, refining language generation, and incorporating error analysis to aid in debugging and model improvement.

Table 14: Summary of types of weakness of model responses in the cause and result QA pairs.

| Weakness Type | Description |
|---------------|-------------|
| Multiple Speculations, Lack of Focus | Lists several possible explanations without identifying the most likely cause |
| Missing Visual Evidence | Fails to reference visual details such as environment or human actions |
| Inaccurate Abnormal Event Detection | Cannot correctly identify which event is the core "abnormal" incident |
| Incomplete Causal Chain | Mentions only the cause or the consequence, without connecting the causal steps |
| Overgeneralized Language | Uses vague or abstract expressions instead of specific reasoning |

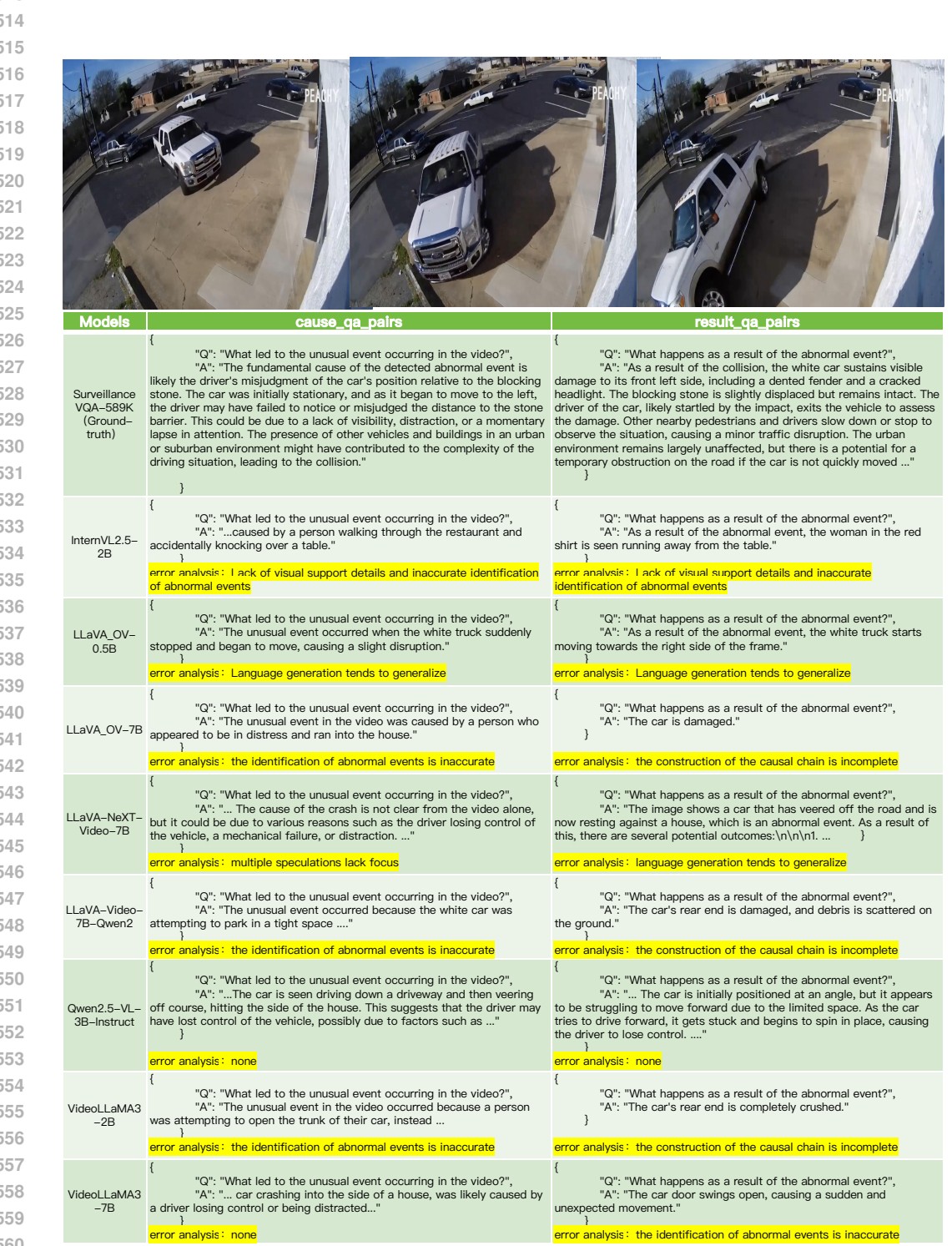

Figure 12: Failed examples on cause QA and result QA types

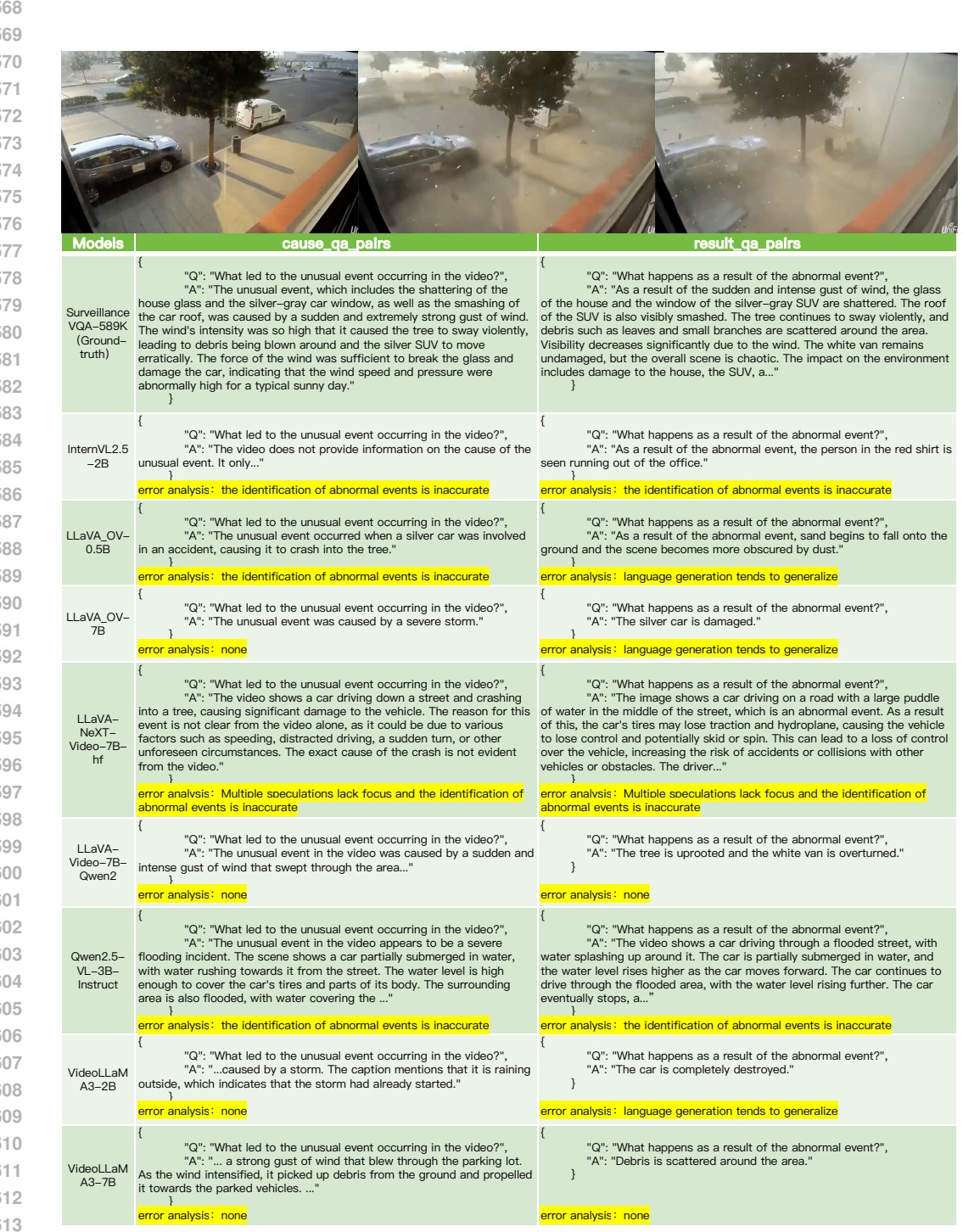

| Models | cause_qa_pairs | result_qa_pairs |
|---|---|---|
| Surveillance VQA–589K (Ground–truth) | { "Q": "What led to the unusual event occurring in the video?", "A": "The unusual event, which includes the shattering of the house glass and the silver–gray car window, as well as the smashing of the car roof, was caused by a sudden and extremely strong gust of wind. The wind's intensity was so high that it caused the tree to sway violently, leading to debris being blown around and the silver SUV to move erratically. The force of the wind was sufficient to break the glass and damage the car, indicating that the wind speed and pressure were abnormally high for a typical sunny day." } | { "Q": "What happens as a result of the abnormal event?", "A": "As a result of the sudden and intense gust of wind, the glass of the house and the window of the silver–gray SUV are shattered. The roof of the SUV is also visibly smashed. The tree continues to sway violently, and debris such as leaves and small branches are scattered around the area. Visibility decreases significantly due to the wind. The white van remains undamaged, but the overall scene is chaotic. The impact on the environment includes damage to the house, the SUV, a..." } |
| InternVL2.5 –2B | { "Q": "What led to the unusual event occurring in the video?", "A": "The video does not provide information on the cause of the unusual event. It only..." } 
 error analysis：the identification of abnormal events is inaccurate | { "Q": "What happens as a result of the abnormal event?", "A": "As a result of the abnormal event, the person in the red shirt is seen running out of the office." } 
 error analysis：the identification of abnormal events is inaccurate |
| LLaVA_OV– 0.5B | { "Q": "What led to the unusual event occurring in the video?", "A": "The unusual event occurred when a silver car was involved in an accident, causing it to crash into the tree." } 
 error analysis：the identification of abnormal events is inaccurate | { "Q": "What happens as a result of the abnormal event?", "A": "As a result of the abnormal event, sand begins to fall onto the ground and the scene becomes more obscured by dust." } 
 error analysis：language generation tends to generalize |
| LLaVA_OV– 7B | { "Q": "What led to the unusual event occurring in the video?", "A": "The unusual event was caused by a severe storm." } 
 error analysis：none | { "Q": "What happens as a result of the abnormal event?", "A": "The silver car is damaged." } 
 error analysis：language generation tends to generalize |
| LLaVA– NeXT– Video–7B– hf | { "Q": "What led to the unusual event occurring in the video?", "A": "The video shows a car driving down a street and crashing into a tree, causing significant damage to the vehicle. The reason for this event is not clear from the video alone, as it could be due to various factors such as speeding, distracted driving, a sudden turn, or other unforeseen circumstances. The exact cause of the crash is not evident from the video." } 
 error analysis：Multiple speculations lack focus and the identification of abnormal events is inaccurate | { "Q": "What happens as a result of the abnormal event?", "A": "The image shows a car driving on a road with a large puddle of water in the middle of the street, which is an abnormal event. As a result of this, the car's tires may lose traction and hydroplane, causing the vehicle to lose control and potentially skid or spin. This can lead to a loss of control over the vehicle, increasing the risk of accidents or collisions with other vehicles or obstacles. The driver..." } 
 error analysis：Multiple speculations lack focus and the identification of abnormal events is inaccurate |
| LLaVA– Video–7B– Qwen2 | { "Q": "What led to the unusual event occurring in the video?", "A": "The unusual event in the video was caused by a sudden and intense gust of wind that swept through the area..." } 
 error analysis：none | { "Q": "What happens as a result of the abnormal event?", "A": "The tree is uprooted and the white van is overturned." } 
 error analysis：none |
| Qwen2.5– VL–3B– Instruct | { "Q": "What led to the unusual event occurring in the video?", "A": "The unusual event in the video appears to be a severe flooding incident. The scene shows a car partially submerged in water, with water rushing towards it from the street. The water level is high enough to cover the car's tires and parts of its body. The surrounding area is also flooded, with water covering the ..." } 
 error analysis：the identification of abnormal events is inaccurate | { "Q": "What happens as a result of the abnormal event?", "A": "The video shows a car driving through a flooded street, with water splashing up around it. The car is partially submerged in water, and the water level rises higher as the car moves forward. The car continues to drive through the flooded area, with the water level rising further. The car eventually stops, a..." } 
 error analysis：the identification of abnormal events is inaccurate |
| VideoLLaM A3–2B | { "Q": "What led to the unusual event occurring in the video?", "A": "...caused by a storm. The caption mentions that it is raining outside, which indicates that the storm had already started." } 
 error analysis：none | { "Q": "What happens as a result of the abnormal event?", "A": "The car is completely destroyed." } 
 error analysis：language generation tends to generalize |
| VideoLLaM A3–7B | { "Q": "What led to the unusual event occurring in the video?", "A": "... a strong gust of wind that blew through the parking lot. As the wind intensified, it picked up debris from the ground and propelled it towards the parked vehicles. ..." } 
 error analysis：none | { "Q": "What happens as a result of the abnormal event?", "A": "Debris is scattered around the area." } 
 error analysis：none |

Figure 13: Failed examples on cause QA and result QA types

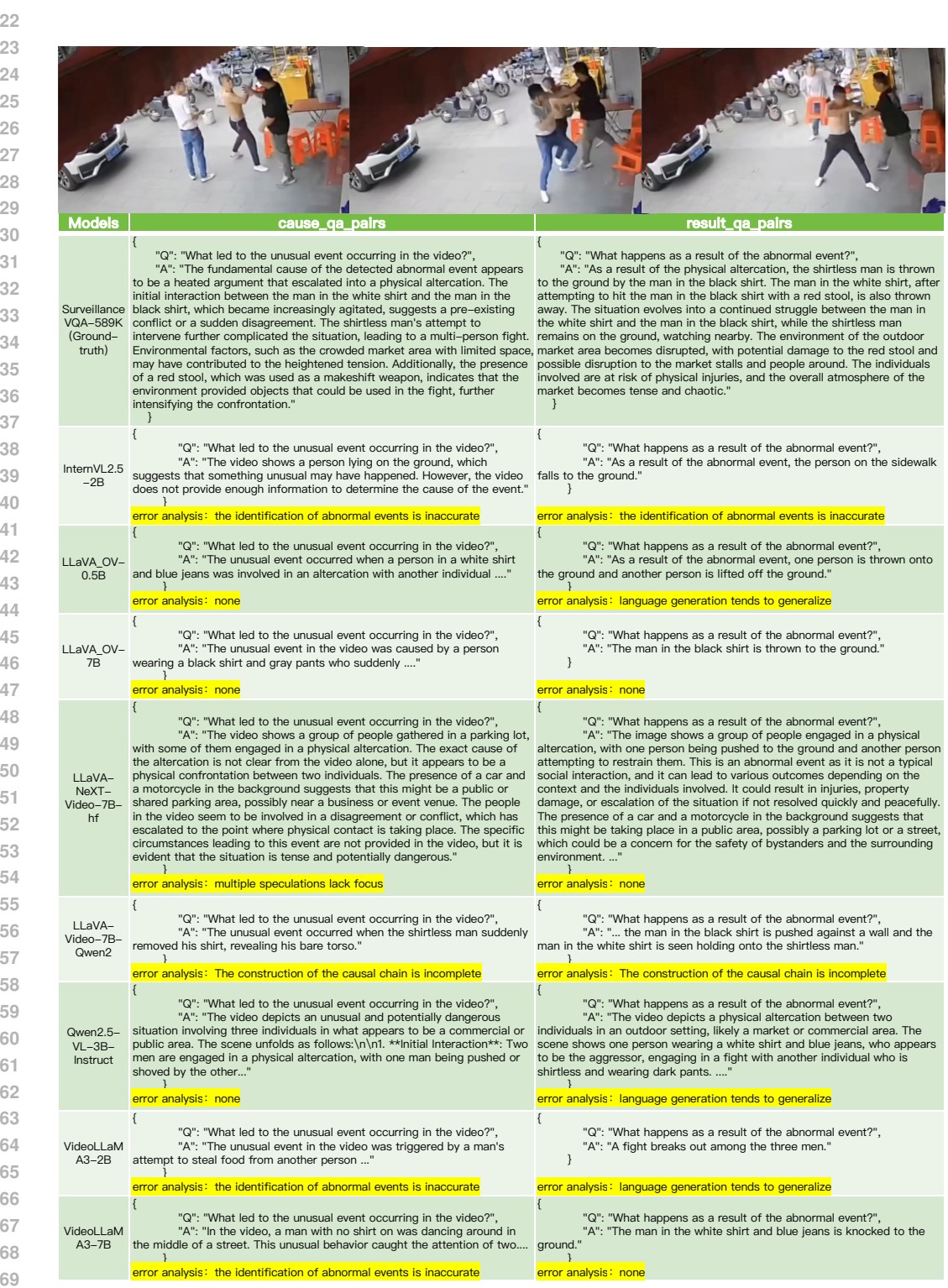

Figure 14: Failed examples on cause QA and result QA types

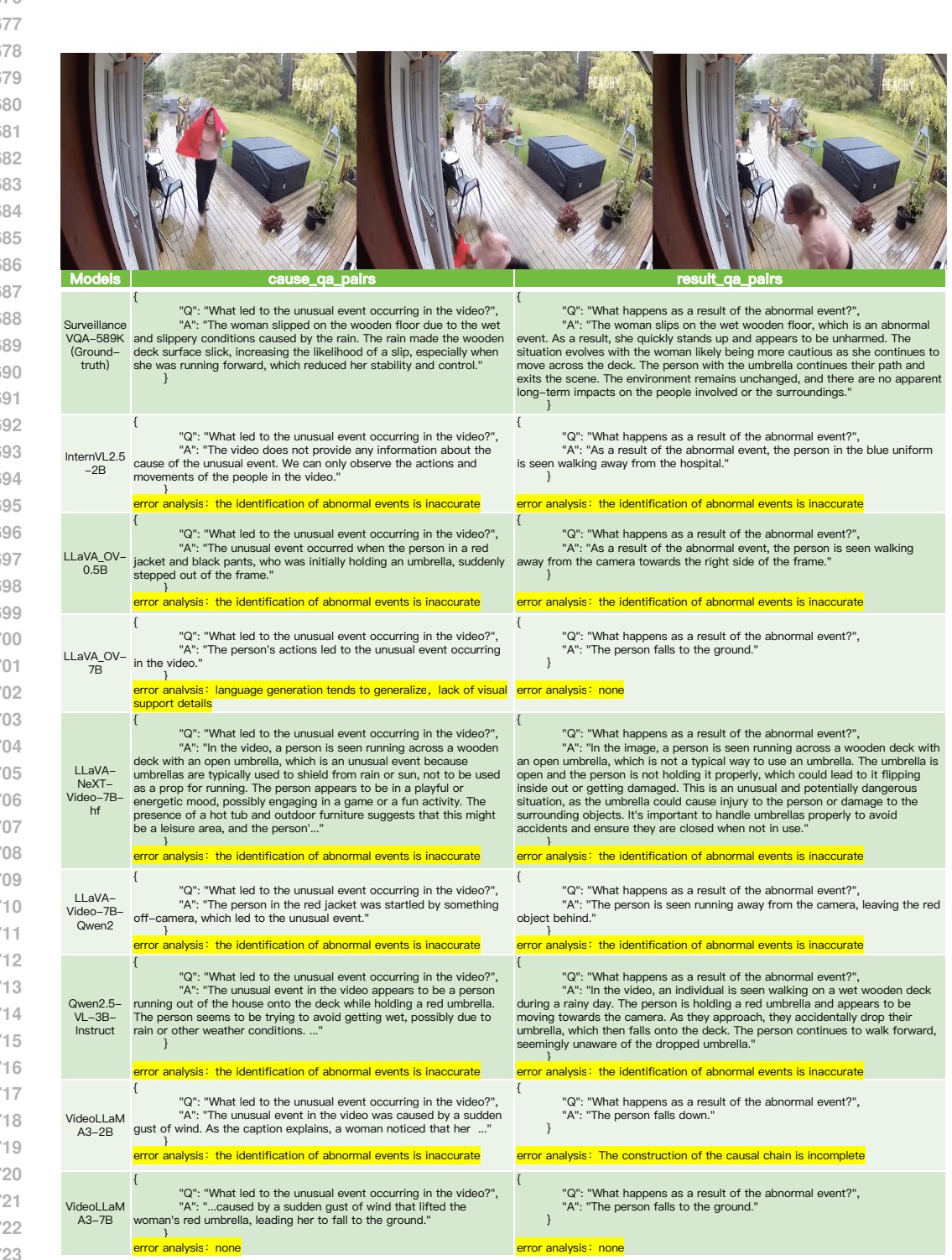

Figure 15: Failed examples on cause QA and result QA types

## G    OTHER CONSIDERATIONS

### G.1    DATASET QUALITY ANALYSIS

Here, we analyze the dataset quality, showing that the measures we have taken can provide a guarantee for it.

First, as detailed in Section 3.1, our QA generation pipeline incorporates both human-labeled video descriptions and LLaVA-Video outputs, thereby combining human intuition and LLM generalization. The human-labeled descriptions produced through a rigorous process, involving detailed guidelines, multiple rounds of expert review, and cross-verification, serve as a strong anchor for grounding the QA generation in real-world semantics and narrative complexity.

Second, using human-labeled video descriptions and LLaVA-Video outputs, we further constructed QA pairs by prompting LLMs. We designed a diverse set of 12 QA types, each with carefully crafted and repeatedly tested prompt templates. This was done to reduce prompt-induced bias and to ensure coverage of a wide range of cognitive demands, including causal reasoning and temporal/spatial understanding.

Third, we conducted a systematic human evaluation after generation. Specifically, 5% of the QA pairs across all types were randomly sampled and assessed by a panel of three AI researchers. Each pair was rated by two independent reviewers based on video verification, semantic accuracy, completeness, and linguistic clarity. The results showed high inter-rater agreement, all the QA pairs were qualified, and 90.3% of the QA pairs were deemed to be of high quality. This post-hoc human validation provides an important counterbalance to the potential limitations of LLM-only generation.

Finally, we believe that our hybrid approach (human-labeled annotations, the iterative design and testing of prompt templates, and rigorous human validation), anchoring generation in human-labeled descriptions and including thorough validation, helps to provide a guarantee for the quality of the dataset.

### G.2    LIMITATIONS

However, this study also has some limitations. Model evaluations are based on a limited dataset, and performance may vary when applied to different surveillance environments or low-quality video data. Although larger models perform well in terms of accuracy, they have high computational demands, which may make them challenging to deploy in resource-constrained environments. Future research should focus on optimizing the temporal reasoning capabilities of smaller models and further explore domain-specific fine-tuning techniques to improve performance in various monitoring scenarios.

## H    USAGE OF LLM

In the process of constructing the dataset, we have used multimodal large models and large language models as automation tools, with detailed explanations provided in the relevant sections. Additionally, large models were also employed for language modification and refinement to present the content in a more readable manner.

