# OpenReview forum: "SurveillanceVQA-589K: A Benchmark for Comprehensive Surveillance Video-Language Understanding with Large Models"
_ICLR.cc/2026/Conference — ICLR 2026 Conference Withdrawn Submission_

### Official Review · Reviewer_DCiy · 2025-10-16

**Soundness:** 2
**Presentation:** 2
**Contribution:** 2
**Rating:** 4
**Confidence:** 2

**Summary:**

This paper addresses the underexplored area of surveillance video understanding within the vision-language research community. To this end, the authors introduce SurveillanceVQA-589K, a large-scale open-ended video question answering (VQA) benchmark specifically designed for the surveillance domain. The dataset is constructed through a semi-automated pipeline that combines human-written captions with outputs from large vision-language models. It encompasses diverse reasoning dimensions, including temporal reasoning, causal inference, spatial understanding, and anomaly interpretation, covering both normal and abnormal surveillance scenarios. Using this benchmark, the authors demonstrate that existing vision-language models exhibit significant limitations in handling real-world surveillance tasks, particularly those involving causal reasoning and anomaly detection.

**Strengths:**

1. The paper is clearly written and well-organized.
2. The proposed SurveillanceVQA-589K benchmark seems to extend beyond simple descriptive tasks to include higher-level reasoning such as logical reasoning, causal inference, and complex semantic comprehension of video content.
3. The authors perform a comprehensive experimental evaluation using a diverse set of LVLMs, providing valuable insights into their performance and limitations in surveillance-related scenarios.

**Weaknesses:**

1. Although the benchmark includes diverse question types such as causal, temporal, and spatial reasoning, these dimensions have already been explored in prior video understanding benchmarks (e.g., VideoMME [1], MVBench [2]). The main novelty thus lies primarily in the surveillance domain rather than in the reasoning taxonomy itself, which somewhat limits the originality of the contribution. To enhance its distinctiveness, the benchmark could introduce evaluation dimensions specifically tailored to abnormal or security-critical scenarios in surveillance contexts (e.g., predictive reasoning about potential incidents or prevention strategies), rather than directly adapting question types from existing datasets.

2. The quantitative differences among evaluated LVLMs are relatively small, raising concerns about the sensitivity and reliability of the proposed evaluation protocol. Incorporating a subset of structured question types, such as multiple-choice or constrained-answer formats, could improve the robustness and interpretability of performance comparisons across models.

3. The paper lacks a thorough analysis explaining why the fine-tuned models yield modest performance gains compared to their zero-shot counterparts, despite being trained on a substantial number of samples. Simply attributing this to overfitting is insufficient without empirical evidence or diagnostic analysis. A deeper investigation-such as examining data distribution shifts, label noise, or domain mismatch between training and evaluation splits-would strengthen the credibility of this claim and provide more meaningful insights into model behavior.


[1] Video-MME: The First-Ever Comprehensive Evaluation Benchmark of Multi-modal LLMs in Video Analysis

[2] MVBench: A Comprehensive Multi-modal Video Understanding Benchmark

**Questions:**

1. How much time is required to complete the full annotation pipeline?

---

### Official Review · Reviewer_6V5t · 2025-10-20

**Soundness:** 2
**Presentation:** 3
**Contribution:** 2
**Rating:** 4
**Confidence:** 3

**Summary:**

This paper introduces SurveillanceVQA-589K, a large-scale benchmark dataset for Video QA designed for the surveillance domains, with 589K QA pairs about 12 diverse question types. The collection is based on a hybrid annotation pipeline which combines human-written captions with LLM assistance to generate the QA pairs at scale. The paper evaluates 12 different models on this new benchmark, notably showing that current models struggle with complex reasoning, especially in identifying the causes of abnormal events and predicting their results. The authors also show that fine-tuning models on this dataset provides some gains, but doesn't solve the core reasoning challenges and can lead to catastrophic forgetting.

**Strengths:**

- Large scale of this dataset compared to prior work
- Evaluation including both multiple open source and proprietary models

**Weaknesses:**

- Simple baselines like text-only, single-image and human evaluation are not presented and would help assessing the benchmark.
- The fact that fine-tuning on the data does not help much is worrying and could suggest issue in the generated data (e.g. lack of diversity of the generated QAs).

**Questions:**

- Is there any analysis on how much the consolidated caption (or ultimately the generated QA) rely on the human or model caption? e.g. can the QA be answered from one of these two captions?

**Details Of Ethics Concerns:**

This is a paper about surveillance which is a field that can raise some ethical concerns.

---

### Official Review · Reviewer_T6N1 · 2025-10-25

**Soundness:** 2
**Presentation:** 3
**Contribution:** 2
**Rating:** 2
**Confidence:** 4

**Summary:**

This paper introduces a VideoQA benchmark for comprehensive surveillance video understanding. The dataset comprises both normal and abnormal video clips and related questions. The questions are AI generated based on video narratives annotated by human or generated by AI Models. It then provides benchmark results of modern large multimodal models and analyzes their strengths and weakness from multiple aspects based on question categories. The findings point to better causal reasoning as well as more data for finetuning.

**Strengths:**

1.	It constructs a large-scale dataset for comprehensive cross-modal surveillance video understanding. The dataset could be of significant value for social security.
2.	It provides well-classified questions, which would benefit model analysis.
3.	It provides results of SOTA LVLMs, analyze their success and failure points, and gives suggestions for future research.

**Weaknesses:**

1.	The dataset seems to be biased to LLaVA and Qwen series of modes, since LLaVA-Video and Qwen-Max are employed for caption and QA generation respectively. The results in Table 3 and 4 show that LLaVA-Video outperform other opensource models, which further confirms my concern.
2.	The QAs are directly generated by Qwen-max, without further human checking for answer correctness. This raises concern about QA quality.
3.	Some basic data statistics should be moved to the main text, for example, what is the size (number of videos and QAs) of the train and test set？
4.	Figure 4 (b), while the title shows QA example, there is no answer for the question.
5.	The evaluation metric is vague. It is unknow how well the judgement aligns with human reviewers.
6.	The current analyses of model behaviors in Table 3 and 4 are unconvincing. The authors ignore an important fact that the QAs are generated by LLaVA-Video and Qwen-Max, which could give strong priors for LLaVA and Qwen series of models during testing. Also, why not provide the results of Qwen2.5-VL 7B? Additionally, a human performance would be helpful to better understand the model behaviors and the soundness of the evaluation metric.
7.	I suggest the authors to provide the average results of each model for better understanding in Table 4.
8.	The suggestion about researching on casual reasoning and domain-adaptive pretraining is not interesting. Many other datasets can support such conclusions. It would be better to summarize more specific challenges and suggestions for surveillance video understanding.

**Questions:**

1.	In Table 4, does finetuning with normal video QA data help abnormal video QA? It would be interesting to see such analyses.
2.	What are the results of API-called models for normal video qa?
3.	Can the benchmarked challenge simply be solved by collecting more data for training? If not, what challenges cannot be well solved?

---

### Official Review · Reviewer_4NkE · 2025-10-28

**Soundness:** 2
**Presentation:** 2
**Contribution:** 2
**Rating:** 2
**Confidence:** 4

**Summary:**

This paper focuses on video understanding in surveillance scenarios and introduces a large-scale benchmark dataset called SurveillanceVQA-589K. The dataset includes over 580K open-ended QA pairs covering both normal and abnormal events, aiming to support semantic and temporal reasoning research in surveillance contexts. The authors use a hybrid of human and llm-generated questions, and evaluate model performance through an llm-based scoring system. Several existing multimodal models are benchmarked to demonstrate the task’s difficulty and provide baseline results.

**Strengths:**

A key strength of this paper is its attempt to extend video question answering into the practically important domain of surveillance. The proposed large-scale dataset and the use of open-ended QA with a hybrid generation strategy reflect a degree of originality in task formulation. The paper is well structured, and the methodology and evaluation framework are clearly described. The experimental section includes multiple mainstream multimodal models, offering a broad view of current model performance in this setting. Given the practical significance of surveillance scenarios, this benchmark has potential impact for both research and applied domains.

**Weaknesses:**

This paper has several notable weaknesses.

First, the originality and incremental contribution are limited. The main differences from existing surveillance QA datasets, such as UCA, lie primarily in scaling up the dataset and changing the evaluation protocol, rather than introducing any fundamental advances in task design or domain modeling.

Second, the open-ended QA format places the task awkwardly between captioning and traditional VQA, failing to provide either the strict semantic alignment of captioning tasks or the controllability and diagnostic power of structured VQA. This ambiguity weakens the scientific rigor and interpretability of the task itself.

Third, the evaluation relies entirely on LLM-as-a-judge, which is inherently subjective and highly sensitive to prompt wording and model versions. This undermines reproducibility and makes it difficult to attribute errors or analyze model capabilities meaningfully.

Fourth, the work does not effectively exploit the distinctive properties of surveillance videos (such as fixed viewpoints, low resolution, small targets, long temporal spans, and sparse anomalies) to inform task or evaluation design. Thus, the claimed “surveillance-specific” aspect remains superficial.

Overall, the contribution is largely an engineering-scale extension rather than a conceptual or methodological breakthrough. Strengthening task definition, grounding the design in surveillance-specific characteristics, and adopting more robust, structured evaluation strategies would significantly improve the work.

**Questions:**

1) Could the authors clarify the fundamental differences between SurveillanceVQA and existing surveillance QA datasets such as UCA? Beyond scaling up and changing the evaluation protocol, is there any substantial innovation in task formulation or domain modeling?

2) Surveillance videos have distinctive properties such as long temporal spans, low resolution, small targets, and fixed viewpoints. Did the authors consider explicitly incorporating these characteristics into task design or evaluation metrics?

3) Since LLM-as-a-judge is inherently subjective, did the authors conduct any prompt sensitivity or repeated scoring tests to assess its stability and consistency?

4) Have the authors considered introducing more structured or partially objective evaluation components to improve reproducibility and diagnostic power?

5) Is there any analysis or ablation study that examines how surveillance-specific factors (e.g., temporal grounding challenges, small-object difficulty) influence QA performance, to demonstrate the task’s unique surveillance characteristics?

There are many benchmark papers nowadays, and I do believe such work plays an important role in advancing the field. However, what matters more to me is whether the authors clearly articulate and deeply understand the relationship between data, scenario characteristics, and the capabilities being evaluated. Simply scaling up without capturing these key aspects is less convincing to me. That said, I genuinely look forward to the authors’ responses and deeper reflections on these essential issues.

---

### Author Response · Authors · 2025-11-13
**Withdraw Statement**

We appreciate the reviewers’ time, evaluation, and feedback on our manuscript. However, after carefully comparing the review comments with the content of our paper, we find that there are numerous evident inconsistencies between the critiques and the research we have presented. To avoid an unproductive rebuttal process and extended discussion, we have decided to voluntarily withdraw our submission.

We would like to briefly clarify the core contributions of our work and how it differs from prior research, such as the UCA dataset. SurveillanceVQA introduces substantial innovations in both task formulation and domain modeling. Unlike UCA, which primarily focuses on text captioning, SurveillanceVQA establishes a comprehensive multi-dimensional QA framework containing 589,380 question–answer pairs, designed to satisfy multi-level cognitive understanding needs ranging from normal surveillance scenarios to anomalous events. The benchmark systematically incorporates 12 types of QA tasks, including spatial reasoning, temporal reasoning, and causal reasoning, thereby constructing for the first time a structured QA evaluation framework tailored to surveillance video understanding.

Furthermore, in building the dataset, we employed manual review and multi-stage quality control procedures, with detailed workflows and statistical analyses provided in the appendix. These measures ensure high data reliability and annotation consistency, offering strong support for the validity of our research conclusions. For these reasons, we believe the reviewers’ concerns regarding data quality and insufficient task innovation do not accurately reflect the actual characteristics of our work.

Given the substantial discrepancies between the review comments and the content of our manuscript, we have decided to withdraw the submission and will further refine and improve the paper in future versions. We remain confident that this research carries unique value for the fields of surveillance video understanding and security-oriented video question answering, and that it will make a meaningful contribution to the development of intelligent surveillance video analysis.

---

### Note · Authors · 2025-11-13

**Comment:**

Given the substantial discrepancies between the review comments and the content of our manuscript, we have decided to withdraw the submission and will further refine and improve the paper in future versions. We remain confident that this research carries unique value for the fields of surveillance video understanding and security-oriented video question answering, and that it will make a meaningful contribution to the development of intelligent surveillance video analysis.

**Withdrawal Confirmation:**

I have read and agree with the venue's withdrawal policy on behalf of myself and my co-authors.